# Peripheral modulation of antidepressant targets MAO-B and GABAAR by harmol induces mitohormesis and delays aging in preclinical models

Reversible and sub-lethal stresses to the mitochondria elicit a program of compensatory responses that ultimately improve mitochondrial function, a conserved anti-aging mechanism termed mitohormesis. Here, we show that harmol, a member of the beta-carbolines family with anti-depressant properties, improves mitochondrial function and metabolic parameters, and extends healthspan. Treatment with harmol induces a transient mitochondrial depolarization, a strong mitophagy response, and the AMPK compensatory pathway both in cultured C2C12 myotubes and in male mouse liver, brown adipose tissue and muscle, even though harmol crosses poorly the blood–brain barrier. Mechanistically, simultaneous modulation of the targets of harmol monoamine-oxidase B and GABA-A receptor reproduces harmol-induced mitochondrial improvements. Diet-induced pre-diabetic male mice improve their glucose tolerance, liver steatosis and insulin sensitivity after treatment with harmol. Harmol or a combination of monoamine oxidase B and GABA-A receptor modulators extend the lifespan of hermaphrodite *Caenorhabditis elegans* or female *Drosophila melanogaster*. Finally, two-year-old male and female mice treated with harmol exhibit delayed frailty onset with improved glycemia, exercise performance and strength. Our results reveal that peripheral targeting of monoamine oxidase B and GABA-A receptor, common antidepressant targets, extends healthspan through mitohormesis.

Disruption of mitochondrial function has been linked to different age-associated pathologies[1,2], such as diminished muscle function[3], insulin resistance and diabetes[4], depression or anxiety[5,6]. In turn, strategies aimed at improving mitochondrial function have been shown to slow aging and reduce the incidence and severity of age-related pathologies[1].

Mitohormetic compounds are defined as mild mitochondrial stressors that trigger a compensatory response coordinated between the nucleus and the mitochondria, resulting in a final improvement in mitochondrial function[1,2]. For this, mitohormetic compounds activate a mitochondria quality control program that relies on the elimination of dysfunctional mitochondria by the induction of a mitochondria-specific type of autophagy, termed mitophagy. Also, an increased ADP/ATP ratio upon energetic stress activates AMPK, a crucial mechanism in the mitochondrial quality control program. Mitohormesis can be induced by genetic means in animal models, such as ablation of sub-units of respiratory complexes[7] or the use of conplastic mouse models[8]. Treatment with compounds that induce a reversible stress in the mitochondria have been shown to improve mitochondrial function by a mitohormetic mechanism. This results in improved diabetes and

✉ e-mail: pablojose.fernandez@imdea.org

muscle function or extended lifespan in *Caenorhabditis elegans*, humans and/or mice, as was shown with the anti-diabetic drug metformin[9,10], the NAD[+] precursors nicotinamide riboside (NR) or nicotinamide mononucleotide (NMN)[11,12], the green tea polyphenols epicatechin gallate (ECG) and epigallocatechin gallate (EGCG)[13], D-glucosamine[14] or the pomegranate-derived urolithin A[15,16]. Also, behavioral interventions such as exercise[7,17], calorie restriction or fasting[18] have been suggested to exert mitohormetic effects.

In this work, we describe the β-carboline harmol as a mitohormetic compound. Harmol simultaneously modulated monoamine oxidase B and GABAAR, two relevant regulators of psychiatric depression and anxiety, and this dual modulation was sufficient to reproduce the harmol-mediated improvement of mitochondrial function. Harmol also activated mitophagy and AMPK pathways in vitro and in vivo. Using animal models, we showed that harmol supplementation extended lifespan of *Caenorhabditis elegans* and *Drosophila melanogaster*. In mice, harmol crossed the blood-brain barrier poorly, but improved glucose homeostasis and insulin sensitivity in a model of diet-induced obesity and diabetes, and delayed neuromuscular degeneration in old mice. Our results reveal a group of compounds that target relevant psychiatric pathways in peripheral tissues and improve whole body metabolism, aging and age-related diseases.

## Results

### Screening platform to identify mitochondria-depolarizing products

We first set up a screening platform for mitohormetic products. Previous reports comparing cell lines showed that the mouse myotube cell line C2C12 had good electron transport capacity and was able to efficiently perform oxidative phosphorylation from a variety of sources[19]. C2C12 muscle precursors can be easily and efficiently differentiated to mature myotubes (Fig. S1a) and have been extensively used to perform high-throughput screenings[20–22]. Also, muscle tissue plays a crucial role in the pathogenesis of type 2 diabetes and it has been proposed as one of the main targets for anti-diabetic strategies[23]. For all these reasons, we chose C2C12 cells for our screening. For the mitochondrial membrane potential reporting system, previous reports had shown that several fluorescent dyes accumulate in the inter-membrane space in a manner proportional to mitochondrial membrane potential[24]. From the several described options, rhodamine derivative tetramethylrhodamine methyl ester (TMRM) was the less toxic to the ETC[25], was specially sensitive to gradual changes in mitochondrial membrane potential, and was well suited for high-throughput screenings[26]. The optimal incubation time to reach a stable TMRM fluorescent signal was determined by treating C2C12 myotubes with two intermediate concentrations of TMRM (15 nM or 25 nM, Fig. S1b and S1c) for 10 min, 60 min or 16 h. Stability of TMRM signal was determined by repeated TMRM measures at 0 min and 40 min. As shown in Fig. S1b and S1c, TMRM incubation for short times (10 min and 60 min) resulted in variability between the first read (0 min) and the second read (40 min). Only incubation with TMRM for 16 h resulted in a stable TMRM signal with time. From this point on, we always incubated C2C12 cells with TMRM for 16 h prior to any treatment. Then, a curve with increasing concentrations of TMRM was performed to find the lowest TMRM concentration that resulted in a sensitive and robust decrease of mitochondria depolarization when treated with the mitochondria depolarizer carbonyl cyanide 4-(tri-fluoromethoxy)phenylhydrazone (FCCP). As shown in Fig. S1d, 15 nM TMRM detected FCCP-mediated depolarization with the same sensitivity as higher concentrations, but 5 nM TMRM failed to do so. Therefore, incubations with 15 nM TMRM for 16 h before the addition of compounds were the chosen conditions for the screening, resulting in very specific and sensitive mitochondria staining as shown in Fig. S1e. We also tested the maximum concentration of the solvent of our libraries (DMSO) that did not elicit significant changes in the TMRM signal. As observed in Fig. S1f, 0.1 % DMSO was the highest concentration with no significant effect on the TMRM signal. Using these conditions, we observed a dose-dependent inhibition of the mitochondrial potential by FCCP (Fig. S1g–h). Importantly, with these conditions, we obtained a Z factor above 0.5 for FCCP concentrations as low as 250 nM (Figure S1i), confirming the robustness of our screening platform.

### Selection of mitohormetic compounds

Using the platform described above, we screened 982 compounds isolated from natural sources such as plants, animals, fungi or bacteria (Supplementary Data 1). We tested each compound in triplicate at a final concentration of 1.3 µg/ml (the concentration used for the screening) dissolved in a final concentration of 0.1% DMSO, and using DMSO at 0.1% as negative control. First, we searched for compounds that induced a short-term mitochondrial depolarization. For this, we measured TMRM fluorescence intensity before the addition of the compounds (time point T0) to define the baseline. We then added the compounds and measured TMRM fluorescence after 30 min (T30) or 60 minutes (T60). Depolarizing compounds elicited a decrease in TMRM signal at either of these times that resulted in a strictly standardized mean difference (SSMD[27]) of SSMD ≤ −2 compared to 0.1% DMSO controls, and were identified as hits. This way, 96 positive hits were selected (Fig. 1a). Next, we searched for compounds that induced a transient, non-toxic mitochondrial depolarization. For this, we retested our 96 hits at T30 and T60, and included a long-term time point at 16 h after treatment. We confirmed a short-term mitochondrial depolarization (T30 and/or T60) below SSMD ≤ −2 in 50 compounds (Fig. 1b). In addition, we selected for reversible depolarizers, one of the characteristics of mitohormetic compounds. For this, cells were kept for another 16 hours with the compounds and TMRM fluorescence was measured again. We selected the 36 hits that induced mitochondrial depolarization after a short treatment of 30 and/or 60 min, but not after a 16-hour treatment (green dots in Fig. 1c, with SSMD ≥ −2), indicating an ability of these compounds to reversibly depolarize the mitochondria.

To select for mitohormetic compounds, we next tested the ability of our reversible depolarizers to improve mitochondrial function. For this, we performed a complete analysis of the mitochondrial function using the Mitostress protocol by Seahorse in differentiated C2C12 myotubes treated with our reversible depolarizers for 16 h. More specifically, we selected the compounds that increased the spare respiratory capacity, a parameter that reflects the ability of mitochondria to cope with stressful situations and has been associated with improved mitochondrial fitness and delayed aging[28]. As a positive control, we used the anti-diabetic drug rosiglitazone, which improves mitochondrial biogenesis and function[29,30]. Rosiglitazone was able to depolarize mitochondria in our C2C12-TMRM model (Fig. S1j), and to increase the spare respiratory capacity measured by Seahorse (Fig. S1k). We identified compounds that had been previously described to reduce mitochondrial potential, such as emodin[31] and mangostin[32] (Fig. S1l and S1m). In addition, we found mitohormetic potential in several members of the family of β-carbolines, such as harmol (Fig. 1d) and harmine (Fig. 1e). Norharmane, from the same family, was identified as a reversible depolarizer (Fig. 1c) and showed a non-significant tendency to increase the spare respiratory capacity in the Seahorse experiment (Fig. 1f). Among all the identified compounds, (Fig. 1g), harmol showed the most robust and reproducible increase in spare respiratory capacity.

### In vitro characterization of the mitochondrial effects of harmol

We next studied the molecular effects of harmol on mitochondria. First, we checked by Seahorse if harmol was acting directly on mitochondrial respiration. For this, we acutely treated C2C12 cells with

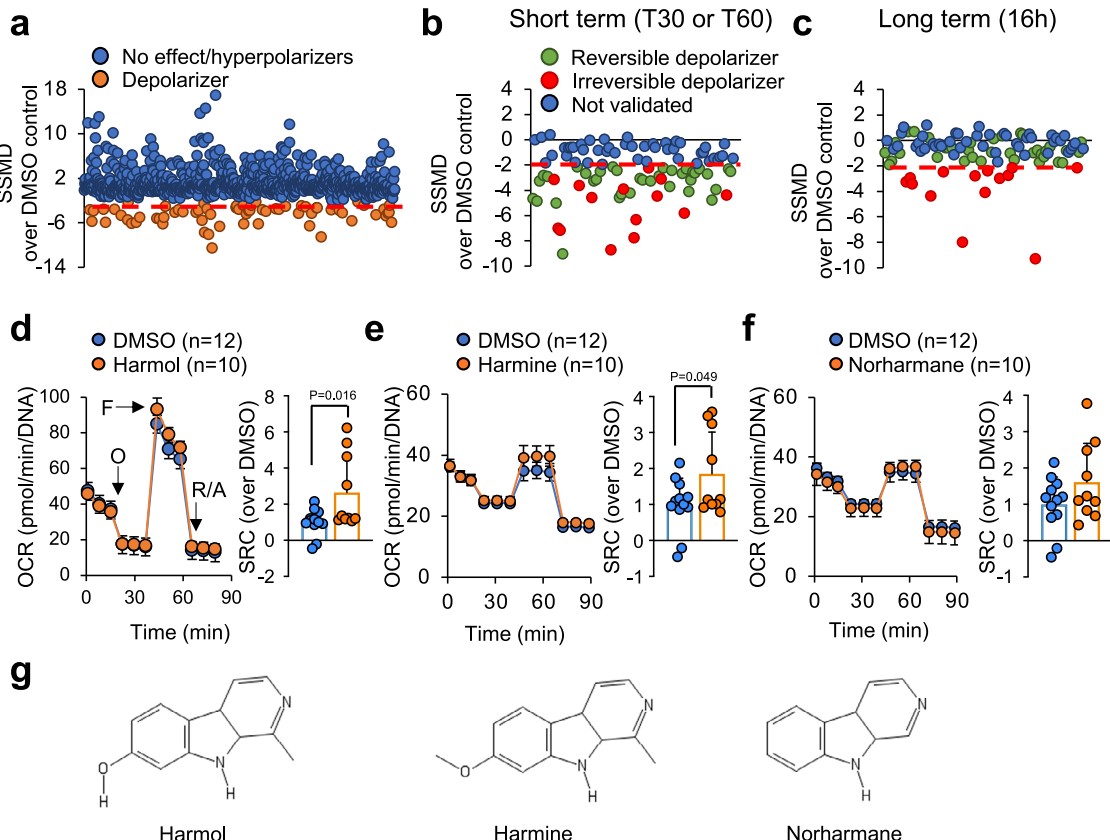

**Fig. 1 | Screening for mitohormetics. a** Strictly standardized mean difference (SSMD) of each of the 982 compounds tested at 30 or 60 min after treatment of differentiated C2C12 myotubes (the lowest SSMD from the two timepoints in indicated). Compounds that induced a SSMD ≤ −2 are shown in orange (depolarizers). Compounds that increased mitochondrial potential or had no effect are shown in blue. **b** SSMD of the 96 depolarizing compounds identified in (**a**) were quantified 30 or 60 min after compound addition, and the lowest SSMD of both times is represented. **c** The same experiment as in (**b**), but measured 16 hours after compound addition. **d–f** Left panels represent the oxygen consumption rates (OCR) in differentiated C2C12 myotubes treated for 16 h with 1.3 μg/ml of harmol

(**d**), harmine (**e**) or norharmane (**f**) following the sequential addition of oligomycin A (O), FCCP (F) and Rotenone/Antimycin A (R/A) at the indicated times (Mitostress Seahorse experiment). Panels on the right represent the spare respiratory capacity (SRC) parameter of the complete Mitostress experiment. **g** Chemical structure of the indicated compounds. Bars and line-connected circles represent the mean of the indicated number of replicates. Individual dots represent independent cell samples. Error bars represent the standard deviation. Isolated circles represent single replicates. Statistical significance was assessed calculating the SSMD parameter (**a–c**) or using the two-tailed unpaired Student t test (**d–f**). P values are indicated when P < 0.05. Source data are provided as a Source Data file.

harmol and measured the immediate oxygen consumption rate (OCR) response using a Seahorse analyzer, reflecting changes in the mitochondrial respiration. As shown in Fig. S2a, acute treatment with harmol, in contrast to rosiglitazone, did not change mitochondrial respiration, indicating that harmol does not act directly on the electron transport chain. Next, we studied the induction of mitophagy and of AMPK, two canonical mechanisms of the mitochondrial quality control program. First, we measured the conversion of the protein LC3-I into its lipidated, autophagosome-associated form LC3-II, a well-described autophagy marker[33]. Interestingly, 1 h after harmol treatment, lipidation of the autophagy marker LC3 was increased in the harmol-treated cells, indicating that harmol treatment activated signs of autophagy (Fig. 2a). We measured the protein levels of the mitophagy marker PINK1[34], and again observed a clear induction in PINK1 levels (Fig. 2a), indicating that harmol triggered autophagy specifically targeting the mitochondria. We also measured the activation of AMPK, another canonical mechanism of the mitochondrial quality control program[35]. We observed that 1 hour of treatment with harmol elicited a moderate, although not statistically significant increase in the phosphorylation of AMPK, similar to the increase observed with the AMPK activator 5-aminoimidazole-4-carboxamide riboside (AICAR). No change was observed in the activity of the AMPK target proteins ACC1 and ACC2 (Fig. 2a and Fig. S2b). We then measured these same markers of the mitochondrial quality control program 3 h after

treatment with harmol. At this timepoint the autophagy and mitophagy markers LC3-II/LC3-I ratio and PINK1, respectively, were not as increased as at 1 h after harmol treatment, although they were still activated when compared to baseline; whereas the AMPK target ACC1 was clearly hyperphosphorylated and activated (Fig. 2b). ACC2, another target of AMPK, was not altered in any of these timepoints (Fig. S2b and S2c).

To better understand the induction of autophagy by harmol, we stained differentiated C2C12 myotubes with a combination of Mitotracker (staining total mitochondria) and Lysotracker (staining lysosomes); or with TMRM (staining active mitochondria, with a detectable mitochondrial potential) and Lysotracker. As shown in Fig. 2c, d, after treatment with harmol we observed a strong co-localization of lysosomes and mitochondria, indicating mitochondria-specific autophagy. Of note, there was almost no co-localization of Lysotracker with TMRM up to 60 min after harmol treatment, indicating that harmol-induced mitochondria-specific autophagy only took place in inactive, TMRM-negative mitochondria (Fig. 2c). To further prove the involvement of autophagy in the mitochondrial depolarization induced by harmol, we treated C2C12 myotubes with chloroquine, that inhibits the fusion of autophagosomes with lysosomes[36]. We observed a dramatic blockage of the harmol-induced mitochondrial depolarization after chloroquine treatment (Fig. 2e and S2d), indicating that mitophagy was essential for harmol-mediated depolarization.

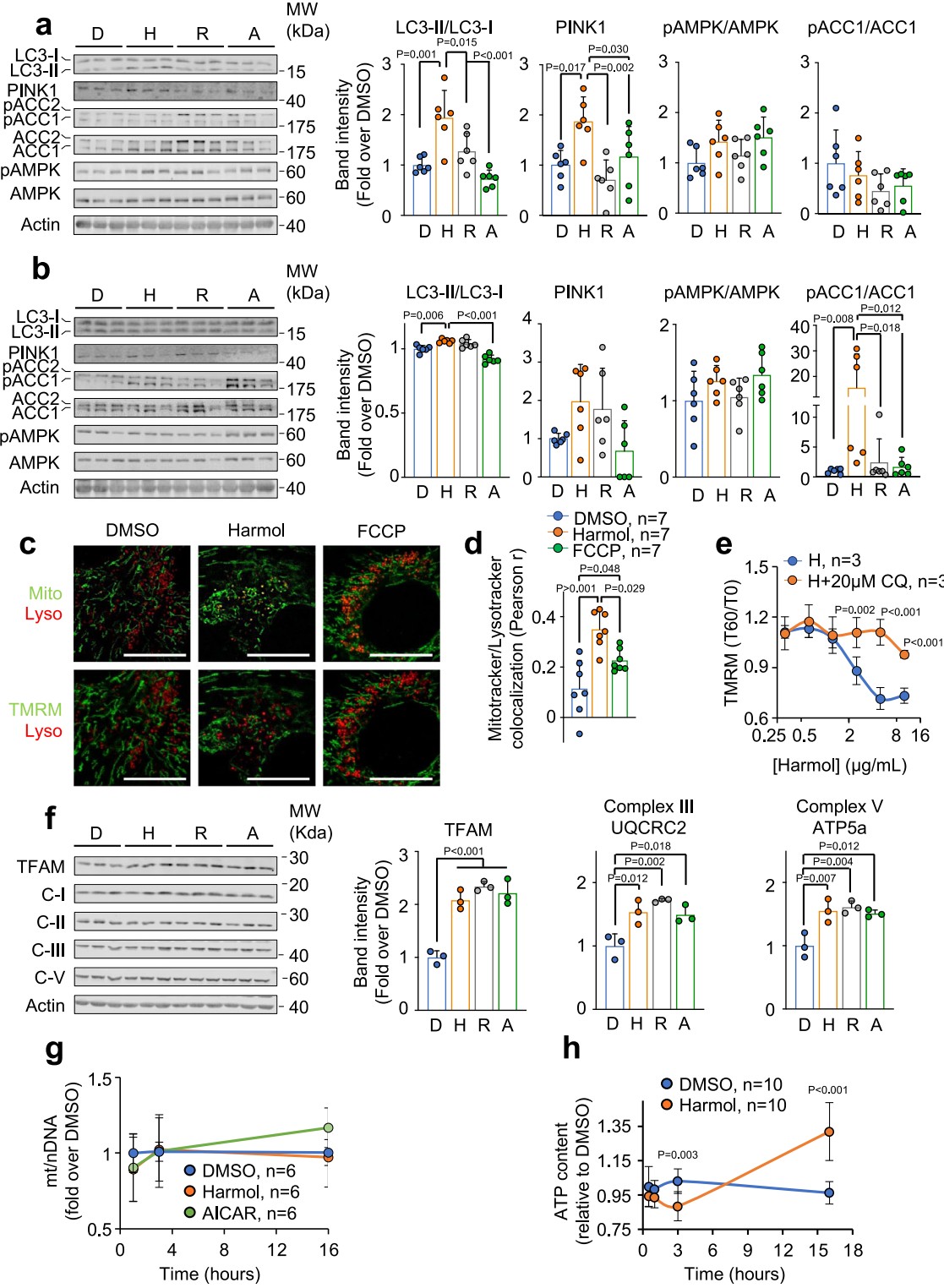

Activation of AMPK and mitophagy leads to a compensatory response that finally results in improved mitochondrial amounts and/or function. To test this mechanism, we measured several markers of mitochondrial function after 16 h of treatment with harmol. We observed a significant increase in the protein expression of the mitochondrial transcription and replication factor TFAM, and of the components of the complex III (UQCRC2) and complex V ATP5a (Fig. 2f and S2e). Regarding mitochondrial content, it remained unchanged during all the treatment, as measured by quantification of the mitochondrial-to-nuclear mt/nDNA ratio, up to 16 h (Fig. 2g). We

finally analyzed ATP production, another illustrative marker of mitochondrial fitness. We observed an initial decrease in the ATP content 3 h after treatment with harmol, in agreement with the induction of the AMPK pathway at this time; and increased ATP levels after 16 h, in agreement with the increased levels of mitochondrial function markers (Fig. 2h).

Finally, we wanted to determine if the observed increase in mitochondrial respiration and function with harmol could be associated to changes in myotube size. For this, we treated differentiated myotubes with harmol. We took photographs of the myotubes at

**Fig. 2 | In vitro mechanistic characterization of the mitochondrial effects of harmol. a, b** Western blots of the indicated proteins in differentiated C2C12 myotubes 1 h (**a**) or 3 h (**b**) after treatment with 0.1% DMSO, 1.3 µg/ml harmol (H), 30 µM rosiglitazone (R) or 250 µM AICAR (A). Quantifications of the significantly different proteins are shown in the bar graphs to the right (*n* = 6 independent cell samples for each condition). **c** Confocal images of differentiated C2C12 cells treated with DMSO (01%), harmol (1.3 µg/ml) or FCCP (500 nM) for 45 min and stained for mitochondria with Mitotracker (Mito); for lysosomes with Lysotracker (Lyso); or for active mitochondria with TMRM. **d** Quantification of the colocalization of Mitotracker and Lysotracker staining, calculated as the Pearson r correlation coefficient in the same cells and treatments as in (**c**). **e** C2C12 myotubes stained with TMRM were treated with the indicated concentrations of harmol alone or combined with 20 µM chloroquine (CQ) for 30 min, and then TMRM intensity was recorded before (T0) and 60 min after (T60) treatment with harmol. **f** Western blots of the indicated proteins in differentiated C2C12 myotubes 16 h after treatment with 0.1% DMSO, 1.3 µg/ml harmol (H), 30 µM rosiglitazone (R) or 250 µM AICAR (A). (*n* = 6 independent cell samples for each condition). **g** Quantitative PCR experiments measuring the ratio between mitochondrial DNA and nuclear DNA (mt/nDNA) in differentiated C2C12 myotubes treated with 1.3 µg/ml harmol or 250 µM AICAR for the indicated times. **h** ATP content in differentiated C2C12 myotubes treated with 0.1% DMSO or with 1.3 µg/ml harmol was assessed at the indicated timepoints (a.u. arbitrary units). Bars and line-connected dots represent the average of the indicated replicates. Dots in bar graphs (**a, b, d, f, g**) represent independent cultured cell samples. Each lane in Western blots (**a, b, f**) represents independent cell samples. Error bars represent the standard deviation. Statistical significance was assessed using the one-way ANOVA (**a, b, d, f**) or the two-way ANOVA (**e, g, h**) tests with Tukey's correction for multiple comparisons. *P* values are indicated when *P* < 0.05. Source data are provided as a Source Data file.

different timepoints, and did not observe any difference in the average myotube size at any timepoint (Fig. S2f).

## In vitro characterization of the mechanisms of harmol-induced mitochondrial response

We next wondered how harmol elicited the mitophagy and mitohormesis responses shown above. For this, we explored the three best-known molecular targets of β-carbolines. First, the β-carboline harmine is a well-known inhibitor of DYRK1A[37,38]. In a recent screening searching for harmine analogs with ability to inhibit DYRK1A in neuroblastoma and glioblastoma cells, harmol was identified as an inhibitor of DYRK1A with low effectiveness against the monoamine oxidase MAO-A (but its effects on the MAO-B isoform was not tested)[39]. To check if harmol effects on mitochondrial depolarization could be caused by DYRK1A inhibition, we treated C2C12 cells with harmol for 5, 10 and 15 min, and checked DYRK1A phosphorylation levels. As shown in Fig. S3a, we did not detect any significant decrease in DYRK1A phosphorylation. These results indicate that inhibition of DYRK1A was not necessary for the mitochondrial depolarization induced by harmol in C2C12 myotubes.

Next, β-carbolines have been shown to inhibit the monoamine oxidase (MAO)[40]. We validated a strong and specific inhibition of the activity of the MAO-B isoform by harmol, starting already at 60 min after treatment and becoming almost totally abrogated 16 h after treatment (Fig. 3a), while the activity of the MAO-A isoform remained unchanged at all times (Fig. S3b). As a positive control, we used harmine, another β-carboline previously described as a pan-MAO inhibitor[41] that we also detected in our initial screen (Fig. 1e). When increasing concentrations of harmol were added together with the MAO-B inhibitor rasagilin, the lowest harmol concentrations tended to depolarize more efficiently than harmol alone, indicating that MAO-B inhibition participated in the depolarizing ability of harmol (Fig. 3b). This cooperation between rasagilin and harmol was significant but not very strong, and was only observed at 3.5 h after the treatment. These results indicated that MAO-B inhibition participates in harmol mechanism of action, but it follows slower kinetics than harmol alone, and it is not sufficient to fully recapitulate harmol functions. MAO enzymes metabolize molecules presenting amine groups, including neurotransmitters as serotonin or dopamine, their best known substrates[42]. In addition, MAO enzymes can also metabolize polyamines, such as putrescine, the metabolic precursor of spermidine[43]. Based on this, we measured alterations in polyamine pools in C2C12 cells treated with harmol. Treatment with harmol significantly increased the levels of the polyamine spermidine (Fig. 3c and Supplementary Data 2).

Since MAO-B inhibition alone did not fully recapitulate all harmol effects, we searched for other possible mechanisms of action. Another well-reported target of β-carbolines is the receptor of the neurotransmitter gamma-aminobutyric acid type A (GABAAR)[44]. To assess if harmol influences GABAAR function, we acutely treated mouse brain slices with harmol or vehicle. We measured spontaneous inhibitory postsynaptic currents (sIPSCs) in hippocampal neurons, which are predominantly GABAergic in the telencephalon and were blocked by the GABAAR antagonist SR95531 (gabazine). We observed that acute application of harmol led to a significant reduction in the frequency of GABAergic currents, and this effect was not observed in vehicle-treated slices (Fig. 3d). Harmol treatment led to a 40% reduction in the frequency in IPSCs but did not affect the amplitude of the remaining sIPSCs observed after harmol application (Fig. S3c). These results suggest that harmol reduced GABAergic neurotransmission but did not completely inhibit GABA binding on GABAAR. To further study the effect of harmol on the GABA receptor, we treated differentiated C2C12 myotubes with the GABAAR antagonist bicuculline or the GABA-receptor inverse agonist FG7142. Neither of these GABAAR modulators elicited any effect in mitochondrial polarization by themselves (Fig. S3d–e). However, when combined with harmol, inhibition of the GABAAR by bicuculline significantly reduced the harmol-mediated mitochondrial depolarization in a dose-dependent manner (Fig. 3e), and strongly blunted the mitophagy induced by harmol (Fig. 3f, g). These results indicated that GABAAR activity is essential for harmol-mediated mitochondrial depolarization. In addition, when the GABAAR inverse agonist FG7142 was added together with very low concentrations of harmol (unable to induce mitochondrial depolarization by themselves), this combination now induced a significant mitochondrial depolarization (Fig. 3h), indicating that harmol activity was potentiated by an inverse agonist of the GABAAR. Finally, treatment of C2C12 differentiated myotubes with a combination of the MAO-B inhibitor rasagiline and the GABAAR inverse agonist FG7142 elicited a mitochondrial membrane depolarization (Fig. 3i), although to a lesser extent than harmol alone. Finally, we measured the effect of MAO-B and GABAAR modulators on the respiratory capacity of C2C12 myotubes. As shown in Fig. 3j and S3f–g, FG7142 and rasagiline alone did not affect the cellular spare respiratory capacity. But, when these two treatments were added together, they induced a synergistic, almost significant improvement in the spare respiratory capacity, recapitulating the effects elicited by harmol. These results indicated that the simultaneous modulation of MAO-B and GABAAR by harmol is at least partly responsible for harmol mitohormetic capacity in C2C12 differentiated myotubes.

Finally, we wondered if harmol effects were dependent on functional mitochondria. To test this point, we treated C2C12 myotubes with vehicle or harmol for 16 hours and measured cellular ATP. In accordance with previous results shown in Fig. 2h, harmol treatment increased ATP production. However, this increase in ATP by harmol was completely blunted if cells were treated with the electron transport chain inhibitors rotenone and antimycin (Fig. S3h), indicating that, indeed, harmol required functional mitochondria to exert its effects.

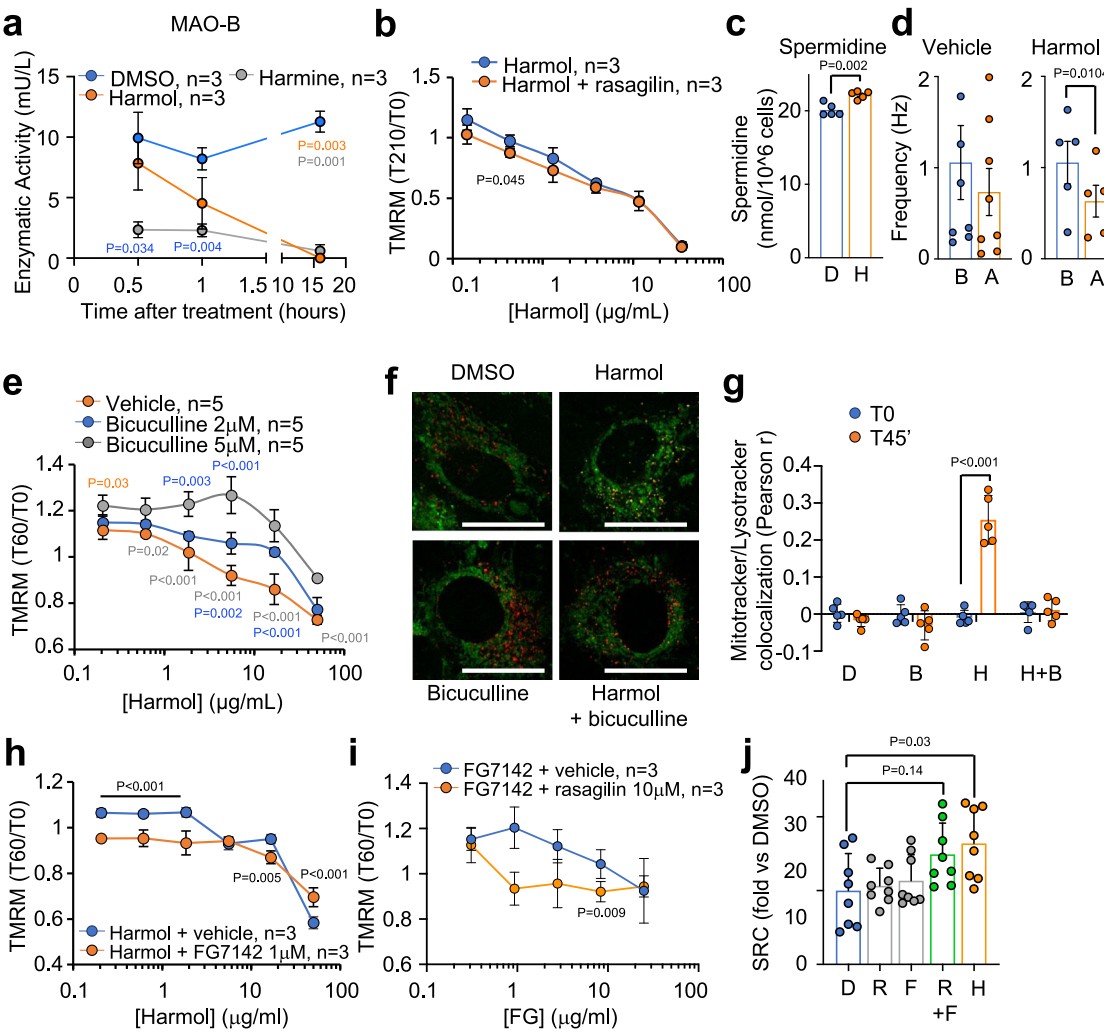

**Fig. 3 | Mechanisms of harmol functions in mitochondria. a** C2C12 myotubes were treated with 0.1% DMSO, 1.3 μg/ml harmol or 1.3 μg/ml harmine, and specific MAO-B activity was measured at the indicate times. **b** C2C12 myotubes stained with TMRM were treated with rasagilin, and then with the indicated concentrations of harmol, and TMRM fluorescence was measured at T0 and T210. **c** C2C12 myotubes were treated with vehicle (0.1% DMSO, D) or with 1.3 μg/ml harmol (H) during 24 h, and spermidine levels were measured. **d** Frequency of spontaneous inhibitory currents (sIPSCs) events measured in mouse hippocampal slices (*n* = 5 from 3 different animals) before (B) and after (A) treatment with 1.3 μg/ml harmol. **e** C2C12 myotubes stained with TMRM were treated with the indicated concentrations of harmol and bicuculline, and TMRM fluorescence was measured at T0 and T60 after treatment. **f** Confocal images of differentiated C2C12 myotubes treated with DMSO (0.1%), harmol (1.3 μg/ml), bicuculline (5 μM) or the combination of harmol + bicuculline, and stained for total mitochondria with Mitotracker (green) and for lysosomes with Lysotracker (red). Size bar = 25 μm. Colocalization of Mitotracker

and Lysotracker is shown in yellow. **g** Quantified Mitotracker and Lysotracker colocalization from (**f**). **h, i** C2C12 myotubes were treated with increasing concentrations of harmol and 1 μM FG7142 (**h**); or with increasing concentrations of FG7142 and 10 μM rasagilin (**i**), and TMRM fluorescence was measured at T0 and T60. **j** Spare respiratory capacity (SRC) of C2C12 myotubes treated for 16 h with 0.1% DMSO, 1 μM rasagiline (R), 2 μM FG7142 (F) or the combination of 1 μM rasagilin and 2 μM FG7142 (R + F), or with 1.3 μg/ml harmol, and then analyzed by Seahorse. Bars and line-connected dots represent the average of the indicated independent replicates. Dots in bar graphs represent independent cell samples. Error bars represent the standard deviation. Statistical significance was assessed using the two-way (**a**, **b**, **e**, **h**, **i**) or the one-way (**g**, **j**) ANOVA with Tukey correction for multiple comparisons; or the two-tailed unpaired Student *t*-test (**c**, **d**). *P* values are indicated when *P* < 0.05. Color codes of *P* values indicate the groups being compared. Source data are provided as a Source Data file.

## In vivo effects of acute treatment with harmol

Next, we treated young, healthy male mice with 100 mg/kg of harmol dissolved in water by oral gavage after removing their food, and kept them in fasting until sacrifice 3 and 7 h later. We took blood samples at different times during this protocol, and did not detect changes in their blood levels of glucose (Fig. S4a). We also measured their levels of the ketone body β-hydroxybutyrate in blood before and after 7 h of fasting. Neither harmol nor rosiglitazone treatments altered the fasting-mediated increase in ketone bodies after 7 h of fasting (Fig. S4b).

We also performed basic pharmacokinetics studies, to determine the bioavailability of harmol with time. As a first in vitro approach, we performed the blood-brain barrier (BBB) specific parallel artificial

membrane permeability assay (PAMPA-BBB), to determine the rate at which harmol crossed an artificial membrane resembling the BBB[45]. As shown in Supplementary Table 1, harmol presented a low permeability to the BBB, indicated by an effective passive permeability coefficient (LogPe) of −4.012, in a scale where LogPe < −4.7 marks totally impermeable compounds, and LogPe > 4.7 marks totally permeable ones. As a reference, we tested other β-carbolines as harmine and norharmane, and the neurotransmitter serotonin, that is totally impermeable to the BBB (LogPe < −4.7). While norharmane presented a similar LogPe to harmol, harmine LogPe was very low (−4.83). This was due to a very high membrane retention, measured by the R parameter: more than 92% of harmine was retained in the artificial lipidic BBB-like membrane, compared with 54.5% of harmol and only

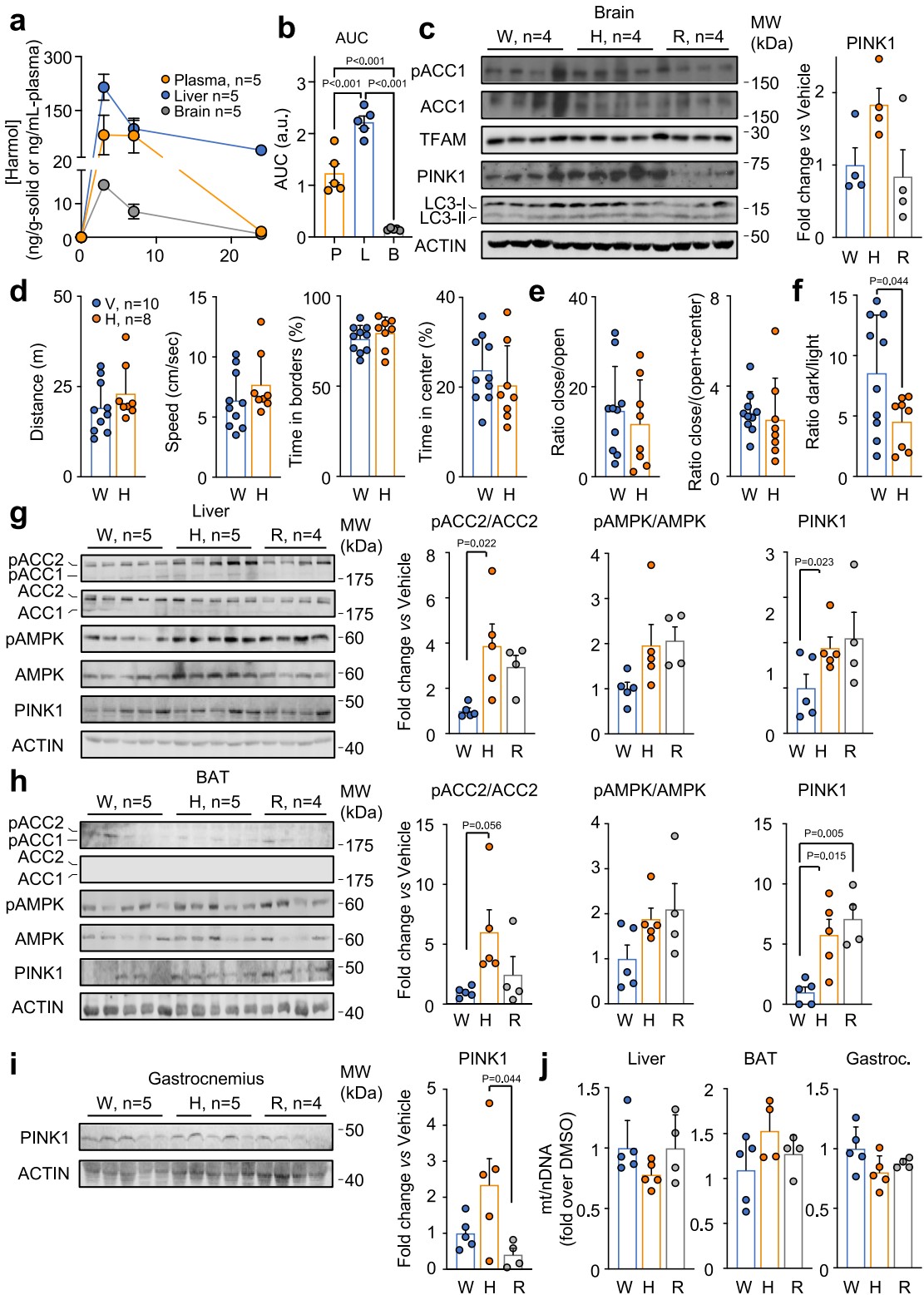

4.8% of serotonin. After equilibrium, the percentage of harmol that had passed from the donor to the acceptor solution through the artificial BBB, measured by the Ca(t)/Cd(0), was 24.64%, similar to norharmane, but not as low as harmine or serotonin. These results indicate that harmol is quite impermeable to the BBB, with <25% crossing it, and is only partially absorbed in the lipid part of the BBB (Supplementary Table 1).

Next, we measured by LC-MS/MS the levels of harmol in plasma, liver and brain samples (for technical details, see Supplementary Table 2) from 12 week-old male mice treated by oral gavage with vehicle or 100 mg/kg harmol for 0, 3, 7 or 24 h. We considered that the shortest time (3 h) would be sufficient for a proper harmol biodistribution; 7 h would be informative about the evolution of harmol with time; and 24 h would indicate how stable harmol levels are in the body. We chose these times based on our results with cultured C2C12 myotubes shown in Fig. 2, where harmol treatment took 1–3 h to exert its full activity. As shown in Fig. 4a and Supplementary Data 3, harmol levels increased very clearly 3 h after treatment with harmol in all the

**Fig. 4 | In vivo effects of acute treatment with harmol. a** Harmol levels were measured by HPLC/MS in plasma, liver and brain samples form 12 week-old C57BL/6JHsdOla male mice treated with 100 mg/kg harmol by oral gavage and sacrificed 3, 7 and 24 hours after treatment. **b** Area under the curve (AUC) of the harmol levels of plasma (P), liver (L) and brain (B) shown in (**a**). **c** Western blots of the indicated proteins in brain 7 h after oral gavage administration of vehicle (water, V), 100 mg/kg harmol (H) or 10 mg/kg rosiglitazone (R). Quantification of PINK1 levels is shown in the bar graph to the right. **d–f** Behavioral assays testing anxiety-like behavior in mice after treatment with 100 mg/kg/day (H) or vehicle (water, W) in the drinking water during 3 weeks: open field test parameters (distance, speed, time in borders and time in center, **d**); elevated plus maze test parameters (ratio close/open and ratio close/(open+center), **e**); and dark-light box test parameter (ratio dark/light, **f**).

**g–i** Western blots of the indicated proteins in liver (**g**), brown adipose tissue (BAT, **h**) or gastrocnemius muscle (**i**) from the same mice shown in (**a–c**). Quantifications of the indicated proteins are shown in the bar graphs to the right. **j** Quantitative PCR experiment measuring the ratio between mitochondrial and nuclear DNA (mt/nDNA) in the indicated tissues from the same mice as shown in (**g–i**). Bars and line-connected dots represent the average of the indicated replicates. Dots in bar graphs (**b–j**) represent samples or measures from independent animals. Each lane in Western blots (**c**, **g–i**) represents samples from independent animals. Error bars represent the standard error of the mean. Statistical significance was assessed using the one-way ANOVA test with Tukey's correction for multiple comparisons (**b**, **c**, **g–j**) or the unpaired two-tailed Student's $t$ test (**d–f**). $P$ values are indicated when $P < 0.05$. Source data are provided as a Source Data file.

analyzed tissues. Liver reached its maximum harmol levels at 3 h and retained high levels of harmol even 24 h after treatment. Plasma also reached high levels of harmol at 3 h, that remained high after 7 h, and dropped dramatically at 24 h after treatment. In turn, we detected very low levels of harmol in the brain after 3 and 7 h of treatment, ~10 times lower than in plasma and liver (Fig. 4a). These time-course experiments were reflected in the area under the curve, that was much lower in brain than in the other tissues (Fig. 4b). In this regard, the brain-plasma AUC ratio through a time-course measurement is a robust biomarker of the ability of a given compound to cross the BBB[46]. According to our time-course measurements, the brain-plasma AUC ratio of harmol was 0.125 in a scale where <0.1 means BBB impermeability, values between 0.3 and 0.5 mean sufficient BBB permeability, and values > 1 mean free BBB cross ability[46]. These results indicate a poor BBB permeability of harmol, in accordance with our in vitro PAMPA-BBB assay.

We next wondered if the same metabolic program that we observed in the C2C12 myotubes after harmol treatment in vitro also took place in harmol-treated mice. To answer this question, we harvested brain, liver, brown adipose tissue (BAT), and skeletal muscle (gastrocnemius) from the same mice used for the LC-MS/MS analysis. In brain, 7 h after treatment with harmol we detected a tendency to increased levels of the mitophagy marker PINK1, although it did not reach statistical significance; while levels of the phosphorylation of the AMPK target ACC1 or the autophagy marker LC3-II/LC3-I were not affected by harmol treatment (Fig. 4c and S4c). We then wondered if the small fraction of harmol crossing the BBB (Supplementary Table 1 and Fig. 4a, b) that elicited the slight increase in PINK1 levels in the brain (Fig. 4c) might have any behavioral effects. To test this possibility, we treated mice with 100 mg/kg/day harmol for 3 weeks. Harmol- and vehicle-treated littermates were tested in a set of paradigms aimed at assessing anxiety-related behavior in rodents. Mice first explored an open field, a novel environment in which mice are exposed to a mildly aversive open area. We did not see any alteration in the time mice spent in the aversive (center), compared with non-aversive (borders) areas, between the harmol-treated mice compared with the control mice (Fig. 4d). We then subjected mice to an elevated plus maze test, composed of two enclosed arms and two open arms elevated over the floor and generating a higher aversive reaction than in the previous open field test. Still, no difference was observed in the harmol-treated mice compared to the vehicle group (Fig. 4e). Lastly, we put mice in a dark-light box, where mice are free to choose between the exploration of a strongly illuminated open box and a covered dark box. In this case, we observed a significantly reduced ratio of time spent in dark/light boxes in the harmol-treated mice when compared to control (Fig. 4f). These results indicated that harmol treatment had a mild anxiolytic effect, that only became apparent upon a strongly aversive stimulus.

We also measured mitohormetic responses in peripheral tissues of mice treated with harmol, vehicle (water) or our positive mitohormetic control, rosiglitazone. Seven hours after treatment with harmol we observed significant increases in the phosphorylation of ACC2 in liver and BAT, and in the levels of the mitophagy marker PINK1 in liver,

BAT and gastrocnemius, compared with water-treated mice (Fig. 4g–i). We also observed a clear tendency, although not significant, towards increased pAMPK in liver and BAT (Fig. 4g, h). Interestingly, at an earlier timepoint, 3 h after treatment, these changes were not yet apparent (Fig. S4d–f). As observed in C2C12 myotubes, 7 h after harmol treatment the increase in pACC, pAMPK and PINK1 were not reflected in significant changes in the mitochondrial cellular content (Fig. 4j and S4g).

## Chronic treatment with harmol improved glucose homeostasis in diet-induced obese mice

We wanted to explore if chronic harmol-mediated improvement of mitochondrial function could enhance metabolic performance. To test this hypothesis, we generated diet-induced obese (DIO) mice by feeding them with HFD (45% of calories from fat) for 4 months. Next, we treated these mice with 100 mg/kg harmol in their drinking water for 3 months. We also included a positive control group treated with 10 mg/kg rosiglitazone, an anti-diabetic drug with mitohormetic effects[29]. Importantly, mice did not show any sign of toxicity, morbidity or mortality during this time, apart from the previously described body weight gain in the rosiglitazone-treated mice[47] (Fig. 5a). We observed that harmol-treated mice showed a clear tendency to reduced body weight gain compared with rosiglitazone and water controls (Fig. 5a). Although we detected a slight decrease in water intake in the harmol-treated mice, that reached significance at several time points (Fig. S5a), food intake was not affected by harmol and, as previously reported, was increased with rosiglitazone during the first weeks of treatment (Fig. 5b and S5b). We also measured the hematological profile of our mice, and we only detected a minor but significant increase in the variation coefficient and standard deviation of their red blood cell distribution width (RDW-CV and RDW-SD) and a small decrease in their hematocrit (Fig. S5c) in the rosiglitazone-treated mice, compared with vehicle- or harmol-treated mice, always within normal rages. At the end of the treatment, we did not find any evidence of tumor appearance, or any other evident pathology in any of the mice. We measured body composition by dual energy X-ray absorptiometry (DEXA). Harmol-treated mice had a reduced fat mass compared with rosiglitazone-treated mice (Fig. 5c), with no decrease in their lean mass (Fig. 5d). Importantly, harmol- and rosiglitazone-treated mice showed a marked decrease in their fasting blood glucose levels (Fig. 5e), indicating an improved glucose homeostasis. We then performed a glucose tolerance test (GTT), and again observed an improved glucose tolerance in harmol- and rosiglitazone-treated mice (Fig. 5f, g). To determine the effect of harmol on insulin function, we first measured the fasting insulinemia, and observed a clear decrease in harmol-treated mice, although not significant (Fig. 5h). We also measured the Homeostatic Model of Assessment for Insulin Resistance (HOMA-IR), taking into account glucose and insulin levels at fasting, and observed that the HOMA-IR of harmol- and rosiglitazone-treated mice were strongly decreased, compared with water-drinking controls, further validating an improved insulin and glucose homeostasis after treatment with harmol (Fig. 5i). However, when an insulin tolerance

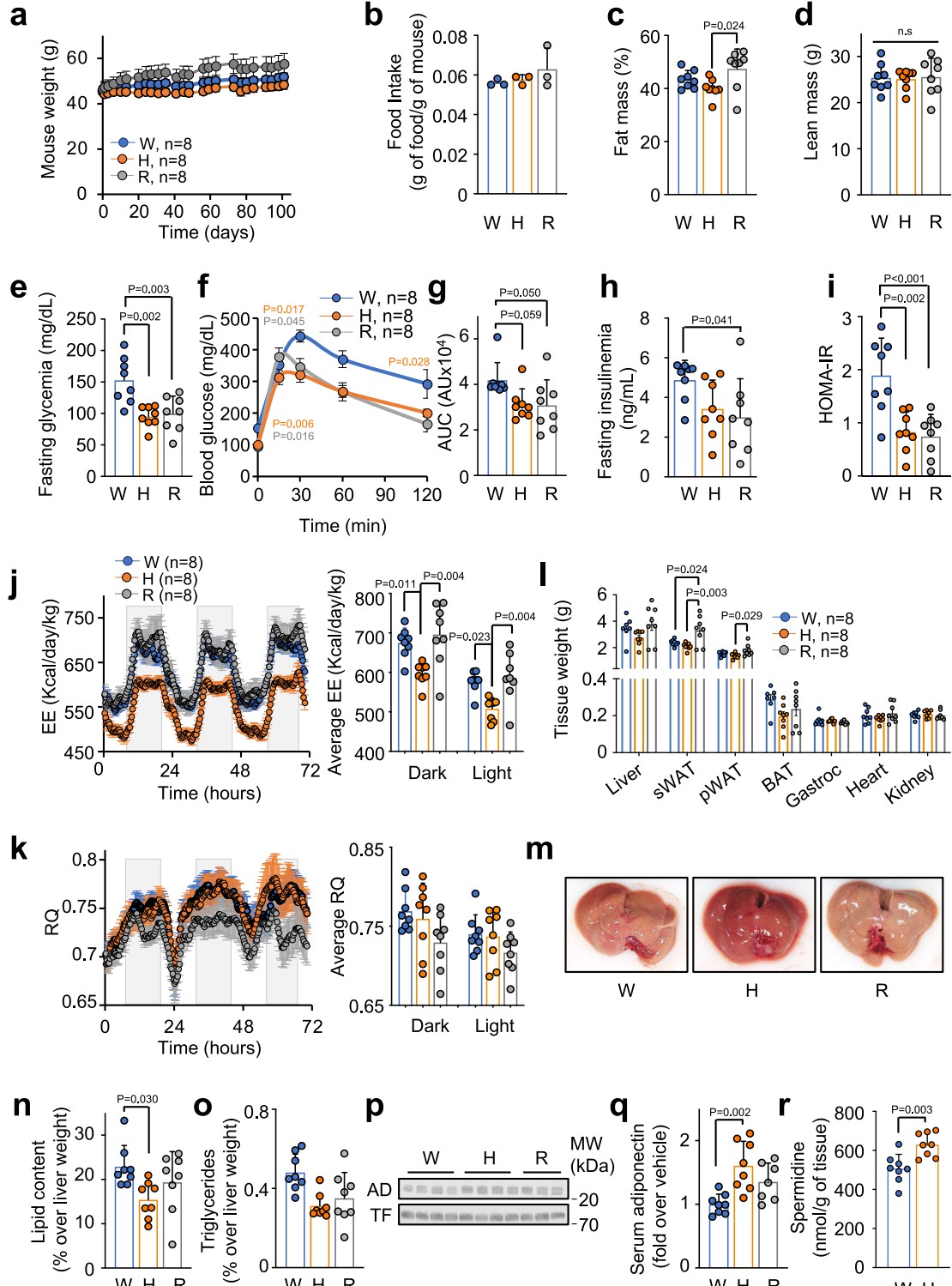

**Fig. 5 | In vivo effects of chronic treatment with harmol. a** Body weight of obese C57BL/6JHsdOla male mice (previously fed with HFD for 4 months since they were 12 weeks old) kept on HFD and treated with vehicle (water, W), 100 mg/kg harmol (H) or 10 mg/kg rosiglitazone in their drinking water for 3 months. **b** Average food intake in the same mice described in (a) during all the time of treatment with the indicated compounds (*n* = 3 cages/condition). At the end of the treatment indicated for (**a**), the same *n* = 8 mice were measured for their fat mass (**c**); lean mass (**d**); glycemia recorded after 16 h of fasting (**e**); glucose tolerance test (**f**; area under the curve is shown in (**g**)); insulinemia (**h**) and HOMA-IR (**i**) after 16 hours of fasting; energy expenditure (EE, **j**) and respiratory quotient (RQ, **k**) with their corresponding average values during day and night shown to the right; weight of the indicated tissues (**l**); liver macroscopic appearance (**m**); lipid and triglyceride content in the liver (**n, o**); blood adiponectin (AD) and transferrin (TF) measured by Western blot (**p**), with the band quantification shown in (**q**); and spermidine levels in the liver (**r**). Bars and line-connected dots represent the average of the indicated number of individuals. Dots in bar graphs (**b–e, g–k, n, o, q, r**) represent samples or measures from independent animals. Error bars represent the standard error of the mean. Statistical significance was assessed using the two-way ANOVA test with Tukey correction for multiple comparisons (**a, f**); and the one-way ANOVA test with Tukey correction for multiple comparisons in panels **b–e, g–l, l**, and **n–r**. *P* values are indicated when *P* < 0.05. Source data are provided as a Source Data file.

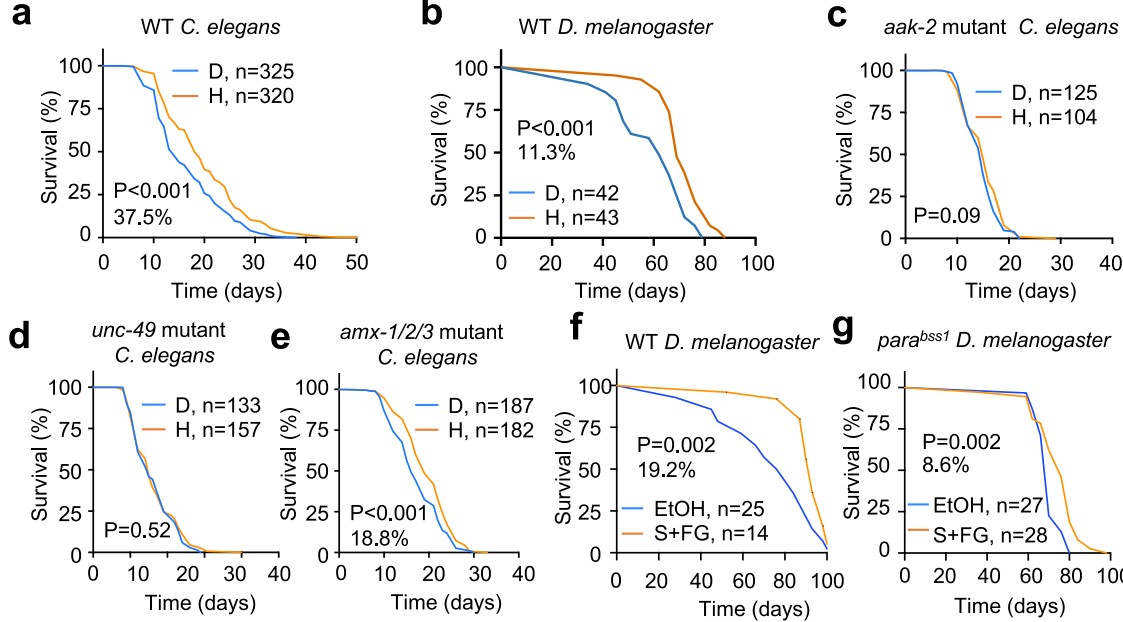

**Fig. 6 | Treatment with harmol extends lifespan in worms and flies.**
**a–d** Kaplan–Meier survival curves including data from at least two independent replicates of the indicated genotypes of hermaphrodite *C. elegans* worms of the N2 strain treated from birth with vehicle (DMSO, D) or with 15 µg/ml harmol (H) (**a**, **c–e**); of the indicated genotypes of Canton-S x w[1118] female *D. melanogaster* flies

(**b**, **f**, **g**) treated with vehicle (DMSO) or 25 µg/ml harmol (**b**); or with vehicle (ethanol, EtOH) or a combination of 0.75 µg/ml selegiline and 10 µg/ml FG7142 (**f**, **g**) since birth. The number of individual flies or worms is indicated in each panel. Statistical significance was assessed using the Log-rank (Mantel-Cox) test. Source data are provided as a Source Data file.

test (ITT) was performed, harmol treatment did not affect insulin sensitivity (Fig. S5d–e).

We also performed an indirect calorimetry experiment with the harmol- and rosiglitazone-treated mice. We observed a remarkable decrease in their energy expenditure (Fig. 5j) and in their VO$_2$ (Fig. S5f), compared with the vehicle-treated control mice, whereas rosiglitazone-treated mice did not show any alteration in these parameters. We assessed the metabolic flexibility of these mice by calculating the Respiratory Quotient (RQ). As shown in Fig. 5k and previously reported[48], rosiglitazone-treated mice showed a significant decrease in their RQ, indicating a stronger usage of lipids than control mice. In turn, harmol-treated mice showed no difference in their RQ values compared with control mice. Of note, neither harmol- nor rosiglitazone-treated mice showed alterations in their locomotor activity (Fig. S5g).

After sacrifice, we measured the weight of several tissues at the time of sacrifice. As shown in Fig. 5l, rosiglitazone-treated mice showed a significant increase in their subcutaneous (sWAT) and perigonadal (pWAT) white adipose depots compared to harmol-treated and/or control mice, as expected from their increased body weight and fat mass shown in Fig. 5a, c. Since DIO mice accumulate lipids in their liver, we wondered if harmol treatment could have a beneficial effect in liver lipid content. Indeed, harmol-treated mice displayed less steatotic livers (Fig. 5m), and accumulated less lipids (Fig. 5n) and triglycerides (Fig. 5o) than control mice. In accordance, liver weight tended to be lighter in harmol-treated mice compared with water-treated mice (Fig. 5l), although this difference did not reach significance. We also measured the plasma levels of adiponectin, an adipokine consistently found downregulated in diabetic individuals in mice[49] and humans[50]. As shown in Fig. 5p, and quantified in Fig. 5q, harmol-treated mice showed a significant increase in their serum adiponectin levels, indicating improved metabolic fitness.

To rule out possible toxicity effects of harmol, we performed a histological evaluation of kidney sections (Fig. S5h), and did not observe any difference between harmol-treated and control mice (Fig. S5i). In addition, blood urea nitrogen (BUN) was not affected by

harmol treatment (Fig. S5j). These results indicate that chronic treatment with harmol did not affect renal function.

To test if increased skeletal muscle mass and/or energy expenditure could be involved in the metabolic effects of harmol, we measured the levels of all isoforms of the muscle protein myosin heavy chain (MHC) in the soleus muscle from mice of the three treatment groups. As shown in Fig. S5k, soleus MHC levels were not changed by the treatment with harmol nor rosiglitazone. Furthermore, we analyzed the muscle fiber type in these soleus skeletal muscles, and we did not detect any change between the treatment groups (Fig. S5l).

Finally, we also measured the accumulation of polyamines in tissues, as the polyamine spermidine was increased in C2C12 myotubes treated with harmol (Fig. 3c), and spermidine has been associated to autophagy-dependent metabolic fitness in the liver[51]. Mice treated with harmol for 3 months accumulated significantly more spermidine in their livers (Fig. 5r and Supplementary Data 2).

**Treatment with harmol extended lifespan in invertebrates**
Mitohormetic compounds have also been shown to delay aging and extend lifespan, a conserved process often reported in invertebrate models such as *Caenorhabditis elegans* and *Drosophila melanogaster*[52]. These organisms have a much shorter lifespan compared to mammalians, so they are ideal to perform this type of experiment. To test if harmol had any effect in the aging process, we treated *Caenorhabditis elegans* worms with harmol since birth and quantified their total lifespan. As shown in Fig. 6a, S6a and Supplementary Data 4a, harmol-treated worms lived significantly longer than vehicle (DMSO)-treated controls, showing a median lifespan extension of 37.5%. In this regard, treatment of *C. elegans* worms with harmol from birth to the end of development slightly decreased phosphorylation of AMPK, although not significantly (Fig. S6b); did not affect the expression of the telomerase catalytic subunit *trt-1* mRNA (Fig. S6c); nor the mitochondrial/nuclear DNA ratio (Fig. S6d). We then tested the lifespan in harmol-treated *Drosophila melanogaster* flies. Again, as shown in Fig. 6b, S6e and Supplementary Data 4b, harmol-treated flies showed a significant lifespan extension of 11.3% compared with control-treated flies. As

observed before with worms and mouse cells and tissues, treatment with harmol did not alter the mitochondrial/nuclear DNA ratio in flies (Fig. S6f). To explore the mechanisms of this harmol-mediated lifespan extension, we measured lifespan in mutant *C. elegans* worms for the AMPK homologue AAK-2. Ablation of *aak-2* in worms completely prevented harmol-mediated lifespan extension (Fig. 6c, S6g and Supplementary Data 4c). Next, we wanted to explore the relevance of harmol target proteins in harmol-mediated lifespan extension. For this, we treated the worm strain with the deletion of the *GABAAR* gene homologue *unc-49* or with the triple deletion of all MAO homologues *amx-1/2/3*. Ablation of the GABAAR homologue *unc-49* completely abrogated harmol-mediated lifespan (Fig. 6d, S6h and Supplementary Data 4d). In contrast, harmol treatment maintained its ability to extend lifespan in the absence of the three MAO homologues *amx-1/2/3*, while treatment with the AMPK inhibitor AICAR actually slightly shortened lifespan (Fig. 6e, S6i and Supplementary Data 4e). These results support our previous mechanistic insights in cultured cells, indicating that harmol inhibits MAO (and, thus, MAO ablation does not affect harmol effects in lifespan); and that a functional GABAAR is necessary for the mitohormetic properties of harmol. Finally, we wanted to determine if this lifespan extension ability of harmol could be reproduced by modulating harmol target proteins pharmacologically, as partly observed in vitro in Fig. 3. For this, we treated *C. elegans* and *D. melanogaster* with the MAO-B inhibitor selegiline, that has been shown to increase lifespan in rats[53], and with the GABAAR inverse agonists FG7142. *C. elegans* worms did not have increased lifespan after treatment with a combination of selegiline and FG7142 (Fig. S6j and Supplementary Data 4f). However, *D. melanogaster* flies treated with this combination lived longer than vehicle-treated flies (Fig. 6f and S6k, and Supplementary Data 4g). We next tested this compound combination in invertebrate models of neurological stress, where MAO and GABAAR modulation may result in more pronounced effects. We first tried an Alzheimer's model, the GMC101 mutant *C. elegans* model[54,55]. We determined viability of these mutant worms by measuring their crawling speed, and worms moving below a threshold were considered non-viable. As shown in Fig. S6l, treatment with harmol resulted in the highest increase in viability in all the days of measurement, although it only reached significance at day 1. In contrast, treatment with selegiline, FG7142 or a combination of these two compounds only increased viability at day 1, but not at the following days. Finally, we also tested flies carrying the bang senseless (*para^bss1*) mutation, that induces epilepsy[56]. When we treated these flies with the combination of selegiline and FG7142, they lived significantly longer than the vehicle-treated flies (Fig. 6g and S6m and Supplementary Data 4h).

### Treatment with harmol delayed frailty onset in aged mice

We next wondered if these age-delaying effects of harmol would also take place in mammals. To test this, we performed the Valencia Score for frailty[57,58] on 2-year-old mice before and after treatment with 100 mg/kg harmol in their drinking water for 2 months. Mice treated with harmol did not show any weight loss, indicating that harmol treatment was safe at this age (Fig. S7a). Recapitulating what we observed in young obese mice (Fig. 5), old mice treated with harmol significantly decreased their fasting glycemia, again pointing to an improved glucose tolerance after harmol treatment (Fig. 7a). Also of interest, mice treated for 2 months with harmol had lower total blood cholesterol (Fig. 7b). Mice treated with harmol showed a clear tendency to maintain their motor coordination for a longer time in the rotarod test (Fig. 7c) and to maintain their grip strength with time (Fig. 7d and S7b), compared with vehicle-treated mice. When muscular strength was tested, harmol-treated mice showed a remarkable improvement compared to vehicle-treated animals, as shown in Fig. 7e, f. When they ran at 75% of their maximal aerobic capacity, harmol-treated mice ran for a longer distance than control-treated mice (Fig. 7g). Cardiorespiratory fitness, as measured by maximal

oxygen uptake (VO$_{2max}$) was significantly higher in the harmol-treated animals when compared with the vehicle-treated ones (Fig. 7h). VO$_{2max}$ is a strong and independent predictor of all-cause and disease-specific mortality in humans[59]. Moreover, at maximum speed, harmol-treated mice produced far less lactate than control-treated mice (Fig. 7i), indicating a later onset of anaerobic glycolysis and better muscular function. We also measured the amount of total myosin heavy chain (MHC) in the soleus muscle from these mice, and we observed increased total MHC levels in harmol-treated mice, more similar to MHC levels in muscle from younger mice (Fig. 7j and S7c), associated with improved muscle function[60]. In addition, we compared the fiber area and fiber type switch in soleus muscle between water- and harmol-treated old mice. We observed that harmol treatment did not affect the muscle fiber size (Fig. S7d–f). However, soleus muscles of mice treated with harmol showed a higher percentage of type II fibers and a lower percentage of type I fibers compared to the control group, a fiber pattern associated to improved muscle function (Fig. S7g–h). When all these parameters were put together and scored according to the Valencia Score, harmol-treated mice scored much lower than control-treated mice (Fig. 7k), indicating that harmol treatment in aged mice protected from frailty.

## Discussion

Improving mitochondrial function is a well-described intervention to delay aging and age-associated diseases[61]. In this work, we have developed a high-throughput screening system based on TMRM staining of C2C12 mouse myotubes and confocal microscopy to efficiently identify bioactive products that induced a transient mitochondrial depolarization. This first screening step was based on previous platforms using TMRM as the cationic dye of choice[62]. Following suggestions reported by Perry and colleagues[24], our screening platform includes additional improvements, such as the avoidance of the quenching effect by optimization of the TMRM concentration; the equilibration of TMRM signal by optimization of the incubation time; and the reading of TMRM signal during several hours, providing a very valuable time-course information. These optimizations in the screening process yielded a sensitive and specific platform, as proven by the identification of well-known mitochondria-depolarizing compounds, such as emodin or mangostin[31,32]. Mitochondria-depolarizing compounds identified using this system were subsequently tested for their ability to increase mitochondrial spare respiratory capacity, a good reporter of improved mitochondrial function[63], using Seahorse technology. By combining these two steps in the same platform, the bioactive products identified with our system fully qualify for mitohormetic products: they induced a transient, non-lethal mitochondrial stress (reduced TMRM signal) that elicited compensatory quality control mechanisms, resulting in an improved mitochondrial function (increased spare respiratory capacity).

Using our screening system, we identified harmol, harmine and norharmane, three members of the β-carbolines family, as bona-fide mitohormetics. Demethylation of harmine to harmol occurs in vivo in humans[64]. β-carbolines are abundant in certain hallucinogen plants, such as *Peganum harmala*, *Banisteriopsis caapi* (also known as "ayahuasca") and *Tribulus terrestris*, and psychedelic as well as antidepressive effects haven been described upon β-carbolines administration[65]. β-carbolines are also present in many foods, most notably coffee beans but also in meat, fish or cereals, as well as in tobacco leaves[66]. Studies about coffee consumption indicated that moderate consumption of coffee was associated with reduced depression incidence[67], better left ventricle systolic and diastolic function[68] and reduced incidence of cardiovascular disease[69]; reduced mild cognitive impairment in aged humans[70]; increase leukocyte's telomere length[71]; and reduced total- and cause-mortality risks[72]. Since coffee is the most abundant source of β-carbolines in human diet, these reports suggest that consumption of β-carbolines at the

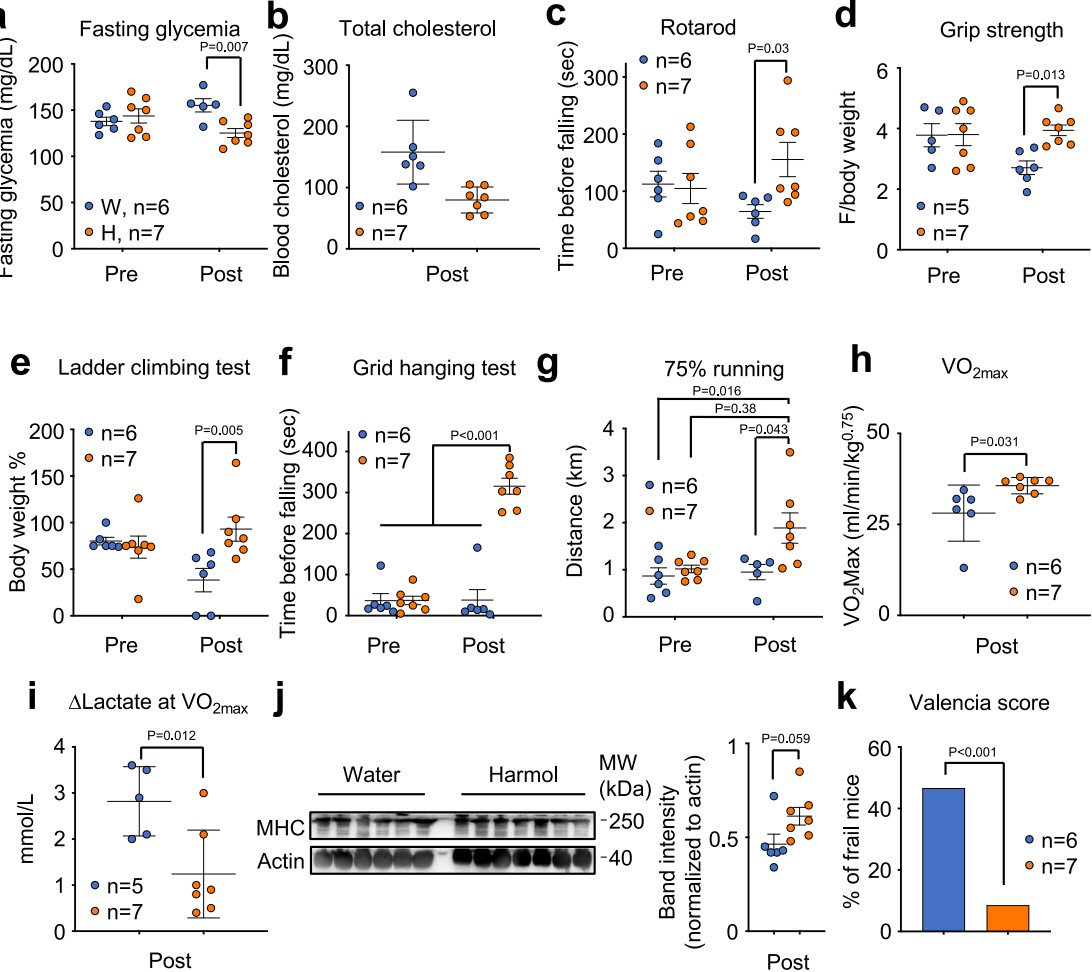

**Fig. 7 | Treatment with harmol delayed aging and reduced frailty in old mice.** (**a**–**f**) Longitudinal assessment of healthspan in 2 year-old male and female C57BL/6JHsdOla mice before (Pre) and after (Post) 2 months of treatment with 100 mg/kg harmol in the drinking water: (**a**) fasting (5 h) glucose levels (n = 5–7); (**b**) total blood cholesterol levels at the end of the intervention (n = 6–7); (**c**) rotarod test (n = 6–7); (**d**) maximal grip strength in grams relativized by animal body weight (BW) (n = 5–7); (**e**) maximal carrying load in the ladder climbing test represented as a percentage of the animal BW (n = 6–7); (**f**) hanging endurance in the grid hanging test (n = 6–7); (**g**) endurance capacity in a running treadmill test (n = 5–7); (**h**) maximal oxygen consumption at the end of the intervention (n = 6–7); (**i**) lactate

increment in the VO₂max test (n = 6–7). (**j**) Western blot against total myosin heavy chain (MHC) in protein extracts from soleus muscle of aged male mice treated with the indicated treatments and normalized to actin. (**k**) Percentage of frail mice (from a total of n = 5–7 at the end of the intervention). Horizontal lines in the dot-plots represent the average of the indicated number of individuals. Dots represent samples or measures from independent animals. Error bars represent the standard error of the mean. Statistical significance was assessed using the two-way ANOVA test with Sidak's correction for multiple comparisons (**a**, **c**–**g**); the unpaired two-tailed Student *t*-test (**b**, **h**–**j**); or the Chi square test (**k**). *P* values are indicated when *P* < 0.05. Source data are provided as a Source Data file.

concentrations present in coffee is, at least, not detrimental for humans. Also, supplementation with the β-carboline harmane was shown to enhance innate immune response and extend lifespan in a model of bacterial infection in *Caenorhabditis elegans*[73].

We focused on harmol, the most potent β-carboline mitohormetic compound from our library. Harmol is a metabolite of harmine, and less toxic than its precursor in experiments with rat liver[74]. To our knowledge, no study focused specifically on harmol treatment has been reported to date. We did not observe any indirect evidence of neurological alterations in harmol-treated mice, as represented in their normal activity after several months of treatment with harmol (Fig. S5g). Harmol crossed poorly the blood-brain barrier (Fig. 4a, b, Fig. S4c, d and Supplementary Table 1). Also, mice treated with harmol did not show significant alterations in their brain in any of the mitophagy markers observed previously in liver, BAT and muscle, with the exception of a slight, non-significant increase in PINK1 levels (Fig. 4c). In addition, we tested the anxiety response to aversive stimuli in mice treated with water or harmol for 3 weeks. We only observed a moderate anxiolytic effect of harmol upon a strong aversive stimulus, such

as a direct and very strongly illuminated area presented in the dark/light box test (Fig. 4f); but no effect was observed in the open field and the elevated plus maze tests, involving milder stressors (Fig. 4d, e). These results indicate that the effects of harmol in the central nervous system are subtle, in accordance with its poor capacity to cross the BBB.

Our results showed that an acute treatment with harmol activated mitophagy and the AMPK pathway, crucial elements of the mitochondrial quality control program, both in cultured C2C12 myotubes and in mouse liver, skeletal muscle and brown adipose tissue; and this activation was much less marked in the brain. Previous reports had already observed induction of autophagy and apoptosis in U251MG glioma cells or in A549 lung adenocarcinoma cells treated with harmol[75,76], although the mechanism responsible for the autophagy induction was not consistent between these two reports. Also, β-carbolines were previously shown to impair mitochondrial respiration in rat hepatocytes, but the exact mechanism was not explored. In addition, harmol was shown to activate the AMPK pathway, that triggered autophagy and mitophagy by phosphorylation of ULK1[77] and

cooperated in the improvement of mitochondrial function observed at later timepoints. In this regard, we have used several markers of autophagy to monitor how harmol regulated it: LC3 lipidation, PINK-1 levels, and lysosome and mitochondria co-localization. However, mitophagy is a complex process that involves a large range of proteins such as Parkin, OPA-1, and many others. A deeper, more thorough study of how harmol affects these pathways would be very valuable. We also show that the initial increase in AMPK phosphorylation after an acute treatment with harmol is followed, after longer treatments, by an improvement in mitochondrial function (Fig. 1d), increased ATP production (Fig. 2h) and a slight reduction in phosphorylation of AAK-2, the AMPK orthologue in *C. elegans* (Fig. S6b). Finally, we did not observe any alteration in the mitochondrial content after harmol treatment in mouse myotubes (Fig. 2g), mouse tissues (Fig. 4j), *C. elegans* worms (Fig. S6d) and *D. melanogaster* flies (Fig. S6f), measured as the ratio between mitochondrial and nuclear DNA. Therefore, the improved mitochondrial function observed with harmol is attained by mechanisms that improve the function of the existing mitochondria, rather than by increasing the total number of mitochondria.

We have also explored the exact mechanisms by which harmol induces mitophagy. β-carbolines are well-known monoamine oxidase (MAO) inhibitors[40] and regulate the activity of other neurotransmitter receptors, most precisely of the GABA receptor GABAAR[44]. As MAO inhibitors, β-carbolines slow down the degradation of MAO-targeted neurotransmitters such as tryptamine, noradrenaline or tyrosine. Regulation of GABA receptors, in turn, is used to treat a wide range of neuronal conditions, ranging from anxiety to depression[78]. MAO inhibitors exert protective effects against neurodegenerative pathologies such as depression or Parkinson's disease[79], and treatment of rodents with β-carbolines elicited anti-depressant-like effects[65], an effect also proposed for GABA receptor regulators[78]. In contrast, blood and cerebellum concentrations of the β-carboline harmane were increased in samples from patients of Parkinson's disease and essential tremor[80,81], and treatment with harmane impaired learning in rats[82], suggesting a potential dose-dependent neurotoxic effect of β-carbolines. Preliminary information, therefore, was not clear about the beneficial or detrimental effects of β-carbolines consumption, although, as commented earlier, harmol crosses the blood-brain barrier with a lower efficacy than harmine.

MAO activities can degrade the polyamine putrescine[43]. In turn, exogenous addition of the polyamine spermidine has been previously shown to induce autophagy and mitophagy, and to extend lifespan in several organisms[83–85]. Therefore, since MAO-B is localized at the mitochondria, MAO-B inhibition by harmol (Fig. 3a) may increase polyamines specifically at the mitochondria, coinciding with the increase in cellular spermidine levels in C2C12 myotubes (Fig. 3c) and mouse livers (Fig. 5r). In turn, increased polyamines may facilitate mitophagy, as observed in C2C12 myotubes (Fig. 2c, d) and in mouse liver, BAT and gastrocnemius (Fig. 4c, g–i) after harmol treatment. Of note, there is no straight homologue of mammalian MAO in *Drosophila*[86]. However, several enzymatic activities substitute for MAO in the fly[87–89], and MAO inhibitors in mammals have been shown to work in flies[90,91]. We therefore hypothesize that the MAO inhibitors, including harmol, used in our work act on the MAO-like activities in our *Drosophila* models, and have an equivalent impact in lifespan extension.

Regarding the effects of harmol on GABAAR, we have found that treatment of brain slices with harmol reduced the frequency of GABAergic synapse events (Fig. 3d); that inhibition in C2C12 cells of the GABAAR with the GABAAR inhibitor bicuculline prevented harmol-mediated mitochondrial depolarization (Fig. 3e) and mitophagy (Fig. 3f, g); and that treatment of C2C12 cells with the reverse agonist FG7142 cooperated with very low doses of harmol to depolarize mitochondria (Fig. 3h). Activation of the GABA receptor has been shown to induce autophagy[92], and GABA receptors have been found localized in the mitochondria[93], possibly close to MAO proteins.

We have identified positive effects of harmol in the homeostasis of glucose and fatty acids (Fig. 5). However, we have not explored the ability of harmol to modify the homeostasis of other metabolites relevant for aging, neuronal modulation and/or frailty. For example, amino acids connect metabolism to neuro-activation, and tryptophan, aspartate and glutamate have substantial effects on the GABA, AMPK, and autophagy pathways. Exploring the role of other metabolites, such as aminoacids, in the beneficial effects of harmol during obesity/diabetes and aging is an interesting field of research.

Physiologically, treatment with harmol significantly extended lifespan in two independent invertebrate models, *C. elegans* worms and *D. melanogaster* flies (Fig. 6). Also, treatment of mice with harmol proved to be safe, and improved glucose tolerance, insulin sensitivity, adipokine profile and liver lipid accumulation in a model of diet-induced obesity and pre-diabetes (Fig. 5), resembling the beneficial effects of metformin, a well-known anti-diabetic mitohormetic used in the clinic. Finally, treatment of old mice (2-year old) with harmol for 2 months was safe and improved a large panel of metabolic and neuromuscular tests, finally yielding a very significant reduction in frailty development (Fig. 7). All these results indicate that harmol treatment is safe in three different species, and shows a very promising bioactivity in diabetes and aging.

Psychiatric and psychologic status have long been associated with healthspan, with mental disorders correlating with shorter lifespan, and higher wellbeing scores associated with longer lifespan[94,95]. However, the molecular mechanisms responsible for these associations are not yet understood. In particular, very little is known about the relationship between psychological and psychiatric status, whole body metabolism and mitochondrial function, crucial determinants of healthspan[96]. In this regard, strategies aimed at improving mitochondrial function have been proposed for the treatment of psychiatric conditions[97]. Our results indicate that harmol may act as an inverse agonist of GABAAR, and that this alteration in the GABA signaling, together with the inhibitory activity of harmol on MAO-B, is necessary for the harmol-induced mitohormesis. Supporting this hypothesis, we observed that the combination of the MAO-B inhibitor rasagiline with the GABA-receptor inverse agonist FG7142 closely resembled harmol functions, depolarizing mitochondria and increasing mitochondrial spare respiratory capacity (Fig. 3i, j). Previous reports showed that treatment with the MAO inhibitor selegiline (also called (-) deprenyl) consistently extended lifespan in several rodent models[53], although the precise mechanism of this extension is not currently known. Also, inhibitors of the GABA receptor have been shown to extend lifespan[98]. We show that the combination of the lifespan-extending MAO-B inhibitor selegiline[53] with FG7142 extended lifespan in WT and *para*^bss1 mutant *Drosophila melanogaster* flies (Fig. 6f, g and S6k, m). It is important to notice that both MAO inhibitors and GABA receptor regulators are clinically used as drugs targeting the central nervous system to treat depression, anxiety or Parkinson's disease in humans. Also, some behavioral interventions that extend healthspan, as exercise or dietary restrictions, target these enzymes[99,100]. Therefore, our results suggest that stimuli that improve psychological status by inhibiting MAO and GABA signaling in the CNS, such as anti-depressants, exercise or calorie restriction, may also induce mitohormesis in the whole body by the mechanism uncovered in this work, leading to better metabolic fitness and improved aging. We propose these mechanisms as an explanation of the long-standing relationship between psychological wellbeing and lifespan[94,95].

Together, we have identified and thoroughly characterized harmol, a type of mitohormetic compound, using our robust high-throughput screening platform, in vitro assays and in vivo experiments with several species. These experiments have allowed us to discover a mechanism for the improvement of mitochondrial function, leading to

potent anti-aging effects in worms, flies and mice. Importantly, harmol shares the molecular targets MAO-B and GABAAR with anti-depressant stimuli, including drugs, exercise or calorie restriction; these results uncover a possible link between antidepressant strategies and improved whole-body metabolic function and healthspan.

## Methods

### Animal experimentation

#### Mouse experiments

**Mouse maintenance.** Animal experiments were conducted at the CNIO (Madrid) and at the University of Valencia, and performed according to protocols approved by the CNIO-ISCIII Ethics Committee for Research and Animal Welfare (CEIyBA) (PROEX 161/18); and by the University of Valencia Ethics Committee for Research and Animal Welfare (License reference: A1444079171882). All the mice used in this work were of C57BL/6JOlaHsd or C57BL/6J backgrounds and, unless otherwise indicated, were fed with the following chow diets: Teklad Global 18% Protein Rodent Diet (Teklad, 2018S) at the CNIO; 2014 Teklad Global 14% Protein Rodent Maintenance Diet (Teklad, 2914) at the Valencia University. All mouse experiments were performed with male mice, except for the 2- year-old experiment where males and females were used. This sex selection was due to availability of mice and cohort homogeneity. Mice were housed at 22 °C, with 12-h light/dark cycles (7:00–19:00) and 55 +/− 15% humidity, and kept in pathogen-free conditions.

**In vivo metabolic assays.** 12-week-old male C57BL/6JOlaHsd mice were rendered pre-diabetic by feeding them with 45% high-fat diet (Research Diets D12451i) for 4 months. For both acute and chronic treatments, harmol and rosiglitazone were dissolved in water. Harmol was purchased from Alfa Aesar (Catalog number: J63827) and rosiglitazone from MedChem Express (Catalog number: HY-17386).

For measurement of glucose, ketone bodies, fasting insulin and adiponectin protein quantification experiments, blood was collected from the tail. Glucose and ketone bodies concentrations were determined using Stat Strip Xpress GLU/KET system (Menarini). For fasting insulin and adiponectin protein quantification experiments, plasma was obtained by centrifuging blood at $1500 \times g$ for 5 min.

For GTT, mice were fasted for 16 hours (from 17:00 to 09:00). Dextrose was dissolved in PBS and injected intraperitoneally at the concentration of 2 g/kg. For ITT, insulin was injected intraperitoneally at the concentration of 0.75 U/kg. In both assays, blood was collected from the tail vein and glucose levels were measured at the indicated times. Fasting insulin quantification was performed using Ultra Sensitive Mouse Insulin ELISA kit (Crystal Chem Inc - 90080). HOMA−IR was calculated using the following equation (HOMA-IR = [(fasting insulin, μU/mL) × (fasting glucose, mg/dl] / 405)[101].

Body composition (fat and lean content) was determined by dual energy X-ray Absorptiometry (DEXA) using a Lunar PIXImus Densitometer (GE Medical Systems). Image acquisition lasted 5 min with mice under anesthesia by inhalation of 2% isofluorane. Image analysis was performed using the PIXImus II Series Densitometers software version 1.46.007 (GEHC).

**Mouse behavioral analysis.** Adult C57BL/6J male mice (3–4 month old) were used for behavioral tests. Mice were maintained under standard housing conditions in 12 h dark/light cycles with food and water ad libitum. Harmol-treated mice were given 1.17 mg/ml harmol in their drinking water ad libitum for 3 weeks (a final dose of 100 mg/kg/day harmol), while control mice were given plain water. All tests were implemented during the light phase with experimenter blind to condition. Mice were handled and habituated to the experimenter for 3 days before the behavioral assays. All behavioral tests were separated at least by 24 h. The order of tests was as follows: open field, elevated plus maze (EPM) and dark-light box. Testing apparatuses were cleaned with a solution of 70% ethanol in water after each trial to avoid olfactory cues. Behavioral tasks were recorded and analyzed as previously described[102].

Open field.

Spontaneous locomotor activity was measured in an open field apparatus. It consisted of a rectangular chamber of $48 \times 48 \times 48$ cm that was made of plastic under uniform light conditions (70 lux). Mice were allowed to explore the arena for 10 minutes. The arena was delimited for analysis into two different regions: center (square area of $30 \times 30$ cm equidistant from the walls) and the remaining borders. The time spent, velocity, distance as well as transitions between the zones was calculated as described in[102].

Elevated plus maze.

The elevated plus maze consisted of four arms ($50 \times 10$ cm) elevated 50 cm above the floor. The plus maze had two closed arms with black acrylic glass walls (30 cm high) and two open (wall-free) arms connected by a central platform. Indirect illumination provided 70 lux to the open arms and 15 lux to the closed arms. Mice were gently placed in the center of the maze and their behavior was recorded for 5 minutes. Time spent and the number of entries in each arm was quantified post hoc[102].

Dark-light box.

The test apparatus consisted of two boxes ($25 \times 25$ cm each) connected by a small aperture: the light box (open box with direct illumination of 450 lux) and the dark box (opaque). Mice were placed facing the dark box and their behavior was recorded for 5 min. Mice were video-tracked and ratio of time in dark box over time in light box was calculated post hoc[102].

**Longitudinal assessment of healthspan in old mice.** Thirteen 23-month-old C57BL/6J mice (9 males, 4 females) were randomly divided in two groups: control ($n = 6$; 4 males, 2 females) and harmol-treated ($n = 7$; 5 males, 2 females). A dose of 100 mg/kg of harmol was dissolved in water and administered to the treated group for 8 weeks. All mice were evaluated before and after the intervention for different functional parameters to assess frailty. Animals were acclimated to all the performed tests with 2 sessions of habituation before the evaluation day.

*Body weight, food and drink intake.* Animals body weight and their food and drink intake were recorded every 7 days using a PB3002 Delta Range balance (Mettler Scales, Toledo, OH).

*Fasting glycemia and total cholesterol.* For the measurement of glycemia, mice were fasted for 5 h (from 7:00 to 12:00); for the measurement of total cholesterol, mice were fasted for 12 h (from 21:00 to 09:00). We collected blood from the saphenous vein and measured both parameters by using a Multicare Lux System (Model 3820001A).

*Motor coordination.* We used the Rotarod (Panlab, Harvard Apparatus) to evaluate motor coordination. The protocol consisted in increasing the speed of the Rotarod from 4 rpm to 40 rpm in 5 min, and the time that the mouse was able to stay on the rod until falling was recorded. Each animal performed three attempts with 5 min of rest between each attempt and the final data recorded was the maximal time achieved in any of the tests.

*Grip strength.* Maximal grip strength was evaluated by using the Grip Strength Meter (Panlab, Harvard Apparatus). The mice were held by the base of the tail by allowing them to grasp the "T" drawbar of the apparatus with their front paws for a few seconds. The peak amount of force applied was automatically registered in grams by the apparatus. Each mouse performed 3 sets of 10 consecutive attempts with 5 min of rest between each set, and the best performance was recorded as maximal grip strength. Since the body weight of mouse influences its force, the grip strength was relativized by dividing the animal weight.

*Ladder climbing test.* We followed a modification of the protocol of Hornberger and Farrar[103] to evaluate maximal strength. Ladder

climbing (1 × 0.18 m, 1 cm grid, 80° incline) was used to assess maximal carrying load with weights secured with adhesive strip to the mouse proximal portion of the tail. The initial climb consisted of carrying a load that was 75% of the animal body weight. Each animal needed ~40 dynamic movements for climbing the entire length of the ladder. When mice reached the top of the ladder, they rested for 5 min. The load was increased by 25% of the mouse body weight in each repetition until the load prevented the animal from climbing the entire ladder. Failure was established after three non-successful attempts to reach at least two-thirds of the ladder. The highest load successfully carried over was considered the maximal carrying load and was represented as a percentage of the mice body weight.

*Grid hanging test*. This is a widely used test that can be considered a more robust assay than grip strength in aging studies because its improvements as a result of geroprotective interventions has been previously demonstrated[104]. A horizontal rectangular grid was used to measure the muscular strength and hanging endurance against the gravitational force with the four limbs of the mouse. The animal was placed on the grid and it was quickly turned upside down. The grid was at a distance of 1 m from the floor with a soft pad underneath to absorb the impact of the fall. We recorded the latency time to fall (in seconds) three times, allowing the mice to rest for 5 min in between. The best time was considered for the statistical analysis.

*Maximal oxygen consumption*. A treadmill for mice with an indirect calorimetry analyzer (Panlab, Harvard Apparatus, Columbus Instruments, Oxylet Pro LE405 gas analyzer) was used for the measurement of their maximal oxygen consumption ($VO_{2max}$, ml/min/$kg^{0.75}$) during an incremental treadmill test. After a warm-up period of 4 min at 6 m/min, the treadmill band velocity was increased by 2.4 m/min every 2 min until exhaustion. Total running time (min) and maximal running speed (m/min) were recorded. The maximal running speed was considered the maximal aerobic workload capacity of the animal[105]. The increase in blood lactate concentration was studied by obtaining blood from the saphenous vein 5 min after the end of the test and comparing with the value at rest.

*Endurance capacity*. From the maximal running speed achieved in the $VO_{2max}$ test, mice performed a continuous running treadmill test to evaluate their endurance. After an incremental warm-up period of 7 min, each mouse ran at 75% of its maximal running speed. Total running distance to exhaustion (km) was registered.

*Frailty assessment (Valencia Score)*. The "Valencia Score" was performed to evaluate frailty in the old animals. This score is based on the previous one for frailty developed for humans by Linda Fried and co-workers[106]. The score includes the measurement of the following five components: unintentional weight loss (change in body weight), low activity level (motor coordination), weakness (grip strength), poor endurance (total running time in the incremental treadmill test), and slowness (maximal running speed in the incremental treadmill test). The frailty score for each age group of animals is calculated as follows: total number of tests failed by the animals at each age group, divided by the total number of tests performed by these animals, expressed in percentage. For more details see refs. 57,58.

**Caenorhabditis elegans experiments.** No ethical approval was required for experiments using invertebrate animals, which have been performed according to EU and national legislation as well as local biosafety regulations. *C. elegans* strains N2 Bristol, GMC101 (*dvIs100 [unc-54p::A-beta-1-42::unc-54 3'-UTR + mtl-2p::GFP]*), *aak-2(ok524)*, RB834 *amx-1(ok659)*, RB1190 *amx-2(ok1235)*, RB1533 *amx-3(ok1838)*, *unc-49(e407)*, and *E. coli* strain OP50 were obtained from the *Caenorhabditis* Genetics Center (CGC, University of Minnesota). *amx-1*, *amx-2*, and *amx-3* mutants were generated by the *C. elegans* Gene Knockout Project at the Oklahoma Medical Research Foundation, part of the International *C. elegans* Gene Knockout Consortium. Strains containing *amx-1*, *amx-2*, or *amx-3* were out-crossed to N2 at least five

times, using PCR to identify the deletions [*amx-1* with p102 (TGA CAA CCG ATG CTT CTC T) and p104 (GAC CGA GTG TGG GGT TAG AG); *amx-2* with p175 (CAG CCT CAA CCA CCT TTT GT) and p204 (GCT GTG CCA ATT CCG ACA C), *amx-3* with p231 (GAA TTC TCG CGC ACG TGA G) and p233 (CTT ATC GCC GAT ATC GTC CG). Then, individual *amx* strains were crossed to generate the *amx-2(ok1235);amx-1(ok659);amx-3(ok1235)* triple mutant. Hermaphrodite worms were routinely grown and maintained on nematode growth media (NGM) agar plates and OP50 at 20 °C as previously described[107].

For lifespan and mobility assays, hermaphrodite worms were age-synchronized using alkaline hypochlorite treatment of gravid adults, and incubated in M9 buffer overnight. This sex selection was due to availability and cohort homogeneity. L1 stage worms were seeded to NGM plates supplemented with compounds, and were treated until death. All compounds were dissolved in dimethyl sulfoxide, and added to media immediately before pouring. Worms were transferred to plates supplemented with 15 μM 5-fluorouracil (Sigma, cat. no. f6627) at the L4 larval stage. 5-fluorouracil treatment continued for the first 2 weeks of life. All assays were performed at 20 °C, and the L4 stage was counted as day 0 of life.

One hundred worms were included per condition in every lifespan experiment. Survival was analyzed every other day and worms were considered dead when they did not respond to repeated prodding. Worms that were missing, displaying internal egg hatching, lost vulva integrity, or burrowing into NGM agar were censored. Statistical analyses of lifespan were calculated by Log-rank (Mantel-Cox) tests on Kaplan-Meier curves in GraphPad Prism 9 for MacOS. All lifespan data and replicates can be found in Supplementary Data 4.

For mobility/viability measurements, worms were grown to the described day of adulthood, and, on the given day, crawling speed was measured and analyzed as previously described[108]. Worms were considered viable when the average speed measured by the wrMTrck plugin for ImageJ was >1.0. Percentage of viable worms was calculated using a custom script in R. Statistical analysis compared each condition to the control using a Fisher Exact test.

For measurements of DNA, RNA, and protein expression, worms were synchronized and grown to L4 larval stage on the described compounds as stated above. Worms were washed from treatment plates, two times in M9 buffer and two times in water, before being snap frozen in liquid nitrogen. Five samples of each condition were grown, each of ~2000 worms.

**Drosophila melanogaster experiments.** No ethical approval was required for experiments using invertebrate animals, which have been performed according to EU and national legislation as well as local biosafety regulations. To avoid fitness defects due to isogeny, our experimental WT individuals were hybrid females of a cross between two reference strains, Canton-CS and w[1118], both obtained from the Bloomington *Drosophila* stock center (BDSC, University of Indiana). For the epilepsy model, we used females of the *para[bss1]* mutant model[56]. This sex selection was due to availability and cohort homogeneity. The food was the standard *Drosophila* cornmeal medium.

From the same cooking batch, we prepared two 96 ml portions. To one of them we added the treatment compounds dissolved in water (harmol) or ethanol (selegiline and FG7142) to obtain the final concentrations, and then dispensed it into 24 tubes, 4 ml per tube. The other portion was dispensed without adding harmol for the group without treatment. *Drosophila* vials producing the progeny were emptied, and virgin female flies that were born overnight were selected on day 0. The flies were divided into two groups: with and without harmol treatment. Three tubes of each genotype with 15 flies per tube were used to perform the survival experiment. Flies were maintained at 25 °C and transferred to a new vial for every 3–4 days until all were dead. Dead flies are easily identified due to their lack of mobility. Fresh food

was made every two weeks. The statistical analyses of lifespan were calculated by Log-rank (Mantel-Cox) tests.

## In vitro experiments

**Cell culture.** Undifferentiated C2C12 myoblasts were purchased from the ATCC (code CRL-1772) and cultured in Dulbecco's modified Eagle's medium (DMEM) containing 25 mM glucose (Lonza, 12-604 F) and complemented with 10% fetal bovine serum (FBS, Tico). Cells were kept at 37 °C in a 5% $CO_2$ atmosphere and passaged every 2 days to prevent differentiation. For myotube differentiation, C2C12 cells were initially seeded in DMEM + 10% FBS. After 16 h, the cells were washed with phosphate-buffered saline (PBS) and the medium was replaced with differentiation medium (DMEM supplemented with 2% Horse Serum (HS)). The medium was replaced every 2 days, and complete differentiation was achieved after 5 days.

For protein and RNA experiments, C2C12 myoblasts were seeded at a density of 200,000 cells/well in six-well plates or 100,000 cells/well in 12-well plates, and the differentiation process was conducted as described previously.

For Seahorse and Western blot experiments, after myotube differentiation, cells were washed with PBS and the medium was replaced with low-glucose differentiation medium (DMEM with 5 mM glucose (Lonza, 12-707F) supplemented with 4 mM glutamine and 2% HS) and incubated for 24 h. The following day, low-glucose media was replaced with fresh medium. All treatments were administered under low-glucose conditions (5 mM).

**Drug screening protocol.** To measure mitochondrial membrane potential in C2C12 myotubes, TMRM (Invitrogen T668) was used as follows. For cell culture, 384 Well CELLSTAR, Black Cell Culture Microplates (Greiner-Bio 781091) were coated with 0.1% gelatin from porcine skin (SIGMA – G1890) for 1 h. Then, 2000 undifferentiated C2C12 myoblasts cells were seeded and differentiated for 5 days. The day before the assay, the differentiation medium was replaced with 99 μL of differentiation medium containing 15 nM of TMRM, and cells were incubated for 16 h at 37 °C in a 5% $CO_2$ atmosphere. Mitochondrial membrane potential was measured using the Opera High Content Screening System (PerkinElmer Opera EvoShell 2.0.0.12199) and analyzed with the Definens Developer XD v2.5 (Definiens) and Acapella 2.6 (Perkin Elmer) software. Basal TMRM fluorescence was measured (T0) prior to treatment. Subsequently, 1 μL of treatment was added and fluorescence was measured at 30 min (T30), 60 minutes (T60), and 16 h after treatment. Pictures were acquired in the same areas throughout the entire experiment, with five pictures taken for each well and an exposure time of 480 ms.

**Colocalization experiments.** Colocalization experiments were conducted using uncoated μ-Slide 8 Well chambers (Ibidi – 80826). Plates were coated with 0.1% gelatin, and 20,000 C2C12 myoblasts were seeded per well. After differentiation, the growth medium was changed to low glucose (5 mM glucose) and 2% horse serum for 40 h. Then, cells were treated with 100 nM Mitrotracker Green (Invitrogen – M7514) to address total mitochondrial mass, 15 nM TMRM for mitochondrial membrane potential, and 25 nM Lysotracker Deep Red (Invitrogen – L12492) to identify lysosomes. The day before the assay, TMRM was loaded, while both Mitotracker and Lysotracker were loaded 4 hours prior to the assay. Colocalization was determined by calculating the Pearson index per pixel for previously detected mitochondrial, TMRM, or lysosome regions, using the Definens Developer XD v2.5 (Definiens) for analysis.

**Seahorse experiments.** Seahorse XF96 Cell Culture Microplates were treated with 0.1% gelatin for 1 h prior to seeding. Gelatin was then removed, and Seahorse wells were washed with PBS. 2500 undifferentiated C2C12 myoblasts were seeded per well and

differentiated for 5 days. After differentiation, the medium was changed to low glucose differentiation medium and incubated for 24 h. Compounds of interest were added and cells were incubated during 16 h. On the day of the assay, the medium was replaced with Seahorse XF base medium (Agilent – 102353-100) supplemented with 10 mM glucose, 1 mM pyruvate, and 2 mM glutamine and cells were incubated for 1 hour at 37 °C without $CO_2$ atmosphere. The Mitostress assay was performed using final concentrations of 2 μM oligomycin (Port A), 300 nM FCCP (Port B), and 1 μM Rotenone/Antimycin A (Port C).

Assay was normalized by total DNA content. Medium was aspirated and cells were lysed with 10 μL lysis buffer (1% SDS, 10 mM EDTA, 50 mM TRIS pH 8.1). 2 μL of the lysate were transferred to a new well containing 23 μL of Hoechst 33342 (SIGMA – B2261) at a concentration of 2.5 μg/mL and fluorescence was immediately measured with a VICTOR Nivo Microplate Reader (PerkinElmer) using an excitation of 355 nm and an emission of 460 nm. To determine mitochondrial function, all parameters were calculated as specified by Agilent. Seahorse data analysis was performed using the Wave 2.4.3 and Microsoft Excel software.

**Western blot.** Mouse tissues and cells were lysed in ice with NP-40 lysis buffer (150 mM NaCl, 1% IGEPAL, 50 mM Trizma base (SIGMA – T4661, pH 8.0) containing 1 mM phenyl-methyl-sulfonyl fluoride (PMSF), 1 mM Sodium Fluoride (NaF), 10 mM sodium orthovanadate (NaOVa) and protease inhibitor cocktail (SIGMA – P8340). Lysates were collected and centrifuged during 15 min at 15,000 × $g$. The supernatant was collected and total proteins were quantified using the DC™ Protein Assay Kit (BioRad – 5000111). Extracts were then diluted to the desired concentration with Laemmli buffer 6X (350 mM TRIS-HCl pH 6.8, 30% glycerol, 10% SDS, 0.6 M DTT and bromophenol blue). Diluted extracts were boiled at 100 °C during 5 min.

Extracts were resolved in SDS-page gels, transferred to nitrocellulose membranes and hybridized in hybridization buffer (TBS-T (20 mM Trizma Base, 150 mM NaCl, 0.01% Tween-20), 5% BSA and 0.02% sodium azide), overnight at 4 °C using antibodies against mouse phospho-AMPKα (Thr172) (pAMPK, 1:250 – 40H9), AMPKα (1:1000 – 2532S), LC3 (1:500 – 2775S), phospho-Acetyl-CoA Carboxylase (Ser79) (pACC, 1:500 – 3661S), ACC (1:1000 – 3676S), PINK1 (1:500 – 6946T) from Cell Signaling; TFAM (1:500 – ab47517), Total OXPHOS Rodent WB Antibody Cocktail (1:1000 – ab110413), Adiponectin (1:500 – ab22554) from Abcam; Transferrin (1:500 – sc-373785) and Total DYRK1A (1:1000 · sc-100376) from Santa Cruz Biotechnology; c-tubulin (1:20,000, GTU-88) or Actin (1:20000 · A1978) from Sigma; Anti-Myosin heavy chain, sarcomere (1:1000 · AB_2147781) from Developmental Studies Hybridoma Bank; and phospho-DYRK1A (1:1000 – 15728122) from Invitrogen. Primary antibodies were incubated with anti-mouse (1:10000 IRDye 800CW, 926-32210, lot number D01110-03) or anti-rabbit (1:10000 IRDye 680RD, 926-68071, lot number D00819-05) secondary antibodies in a mixture of 5% non-fat milk dissolved in TBS-T for 1 h at room temperature. Images were then acquired using LiCor. Band quantification was conducted using ImageJ 1.53n.

**Blood-brain barrier specific parallel artificial membrane permeabilty assay (PAMPA-BBB).** The method published by Muller et al.[45] was followed. Solutions of each compound were prepared in dimethyl sulfoxide (DMSO) at 2 mM and then diluted with 0.01 M PBS (Phosphate Buffered Saline; pH = 7.4) to obtain the donor drug solution with the final nominal concentration of 500 μM in 5% DMSO.

In the acceptor plate (96-well PTFE acceptor plates (Multiscreen Acceptor Plate, MSSACCEPTOR; Millipore)) 300 μL of 0.01 M PBS, 5% DMSO were added. The donor plate (96-well polycarbonate based filter donor plates (Multiscreen™-IP, MAIPNTR10, pore size 0.45 μm; Millipore) was covered with 5 μL of polar brain lipid extract solution

(PBLE), prepared by dissolving 2 mg of PBLE in 20 μL of dodecane and 60 μL of hexane. Afterwards, 150 μL of the 500 μM solution of the test compounds were added over the lipid layer. The donor plate was fitted over the acceptor, covered with wet filter paper and the lid. The system was incubated and shaken for 16 h at 37 °C until equilibrium. The final concentration in the donor plate, acceptor plate and the membrane retention (%MR) were quantified by Nanodrop spectrophotometer 2000 at 245, 242, 235 and 230 nm for harmol, harmine, norharmane and serotonin, respectively. Data were measured in 4 replicates on every single plate. Serotonin was used as negative control.

**Total ATP content determination.** Total ATP content was determined using ATPlite Luminescence Assay System (PerkinElmer) and following the manufacturer's instructions.

**Mitochondrial DNA (mt/nDNA) content measurement.**
a. **DNA Isolation for mt/nDNA.**
Cells and animal tissues were lysed in lysis buffer (1% SDS, 10 mM EDTA and 50 mM Tris-HCl, pH 8.1). Each sample was sonicated to fragment chromatin and liberate DNA. RNA was removed by adding 10 mM EDTA, 35 mM Tris-HCl (pH 6.8) and 35 ng/μL of RNAse. The mix was incubated for 5 min at 37 °C. Samples were further treated with Proteinase K (20 ng/μL) for 2 h at 55 °C. DNA was then extracted using phenol/chloroform extraction. Briefly, 1 volume Acid-Phenol:Chloroform, pH 4.5 (with IAA, 125:24:1) (Invitrogen AM9720) was added and samples were centrifuged for 5 min at 18,500 × g. Supernatant was collected and 1 volume of chloroform was added. After vortexing, samples were centrifuged for 5 min at 18,500 × g. Again, supernatant was collected and treated with a mixture of 300 μM of sodium acetate (pH 5.2), 3 volumes of 100% ethanol and 1 μL of glycogen (Invitrogen – 10814) per 1000 μL of sample volume. DNA was precipitated at −20 °C for 2 h. Samples were centrifuged during 15 min at 18,500 × g and the pellet was washed with ice-cold 70% ethanol. Again, samples were centrifuged at 18,500 × g for 15 min. Supernatant was removed and pellet was allowed to air dry at room temperature. Pellet was solubilized in nuclease-free water and total DNA was quantified using Nanodrop 2000 Spectrophotometer (Thermo Scientific).

b. **mt/nDNA quantification**
For mt/nDNA quantification, quantitative real-time PCR was performed using GoTaq qPCR Master Mix (Promega – A6001) in an ABI PRISM 7900 thermocycler (Thermo Fisher Scientific).

For nDNA and mtDNA, the following primers were used:
Mouse *Ucp2* gene (nDNA):
  – Forward: 5'-CTACAGATGTGGTAAAGGTCCGC-3'
  – Reverse: 5'-GCAATGGTCTTGTAGGCTTCG-3'

Mouse mitochondrial genome (mtDNA):
  – Forward: 5'-ATAACCGCATCGGAGACATC-3'
  – Reverse: 5'-GAGGCCAAATTGTGCTGATT-3'

*C. elegans act-3* gene (nDNA):
  – Forward: GCCTCGTACTGATGGGAAGA
  – Reverse: GCAACTGTGGCAAGATGTGA

*C. elegans nd-1* gene (mtDNA):
  – Forward: 5'-AGCGTCATTTATTGGGAAGAAGAC-3'
  – Reverse: 5'-AAGCTTGTGCTAATCCCATAAATGT-3'

*D. melanogaster RpL32* gene (nDNA):
  – Forward: 5'-AGGCCCAAGATCGTGAAGAA-3'
  – Reverse: 5'-TGTGCACCAGGAACTTCTTGAA-3'

*D. melanogaster 16S rRNA* gene:
  – Forward: 5'-TCGTCCAACCATTCATTCCA-3'
  – Reverse: 5'-TGGCCGCAGTATTTTGACTG-3'

qPCR data was collected and analyzed using an an ABI PRISM 7900 thermocycler with the Sequence Detection Systems SDS 2.4.1 software (Applied Biosystems). Total mitochondrial content was calculated applying the following equation (Mitochondrial content = $2 \times 2^{(CtnDNA-CtmtDNA)}$)[109].

**In vitro electrophysiology in mouse brain slices.** Postnatal day (P) 15–20, mice were deeply anesthetized with isoflurane, decapitated and the brain rapidly removed and placed in ice-cold cutting solution containing (in mM) 248 sucrose, 3 KCl, 0.5 CaCl$_2$, 4 MgCl$_2$, 1.25 NaH$_2$PO$_4$, 26 NaHCO$_3$, and 10 glucose, saturated with 95% O$_2$ and 5% CO$_2$. Coronal slices (300 μm-thick) containing the hippocampus were obtained on a vibratome (Leica VT1000S, Leica Microsystems). Slices were allowed to recover at 37 °C in a submersion chamber containing artificial cerebrospinal fluid aCSF (in mM): 124 NaCl, 3 KCl, 2 CaCl$_2$, 1 MgCl$_2$, 1.25 NaH$_2$PO$_4$, 26 NaHCO$_3$ and 10 glucose saturated with 95% O$_2$ and 5% CO$_2$. For patch clamp recordings in whole cell configuration, slices were transferred to a chamber continuously superfused with aCSF at 23 ± 2 °C on an upright microscope (Nikon) equipped with a Hamamatsu camera to visualize layer allocation (4x objective) and neural somata (40× objective). Borosilicate filamented glass recording electrodes (7–10 MΩ resistance) were prepared from a vertical pipette puller (Narishige PC-100) and filled with intracellular solution containing (in mM): 135 KCl, 10 HEPES, 2 Na-ATP, 0.2 Na-GTP, 2 MgCl$_2$, 0.1 EGTA and adjusted to pH 7.3 and osmolarity (285 mOsm). We used this intracellular solution to measure spontaneous inhibitory postsynaptic currents (sIPSCs) (E$_{Cl}$ ~ 0 mV). sIPSCs currents were isolated by addition of NBQX (10 μM) and APV (40 μM) blocking ionotropic glutamate receptor currents, recorded during 10 min at −70 mV in the absence and presence of harmol (1.3 μg/ml) or vehicle (H$_2$O) and blocked by addition of GABAzine (5 μM). Recordings were acquired with a Multiclamp 700B amplifier, digitized with 1440 A Digidata and analyzed using pCLAMP and MiniAnalysis software.

**Polyamine analysis by mass spectrometry (LC-MS/MS).** Polyamine analysis was performed on whole-tissue or cell extracts. Tissues or cells were snap-frozen in liquid nitrogen and stored at −80 °C until polyamine extraction. For polyamine extraction, 40–50 mg of liver or 4 × 10$^5$ cells were homogenized on ice for 30–60 s using an electrically powered Turrax homogenizer (by IKA) in 1200 μL (tissue) or 600 μL (cells) 5% TCA. 80 μL (tissue) or 100 μL (cells) of this homogenate were immediately mixed with stable-isotope-labeled internal standards and further processed as previously described. A quantitative LC–MS/MS-based determination of polyamines was performed as previously described[110], with minor modifications: after derivatization of TCA extracts, polyamine derivatives were extracted offline by SPE (Strata-X, Polymeric Reversed Phase, 96 well plate). SPE was conditioned with 500 μl acetonitrile, equilibrated with 500 μl distilled water containing 0.2% acetic acid. Derivatized TCA extracts were loaded onto the SPE and after two washing steps with 500 μl of 0.2% acetic acid, samples were eluted with 250 μl 80% acetonitrile containing 0.2% acetic acid. Eluted SPE extracts were subjected to LC-MS/MS (mobile phase: isocratic 80% acetonitrile containing 0.2% acetic acid; flow rate 250 μl/min; HPLC column: Kinetex 2.6 μm C18 100 A 50 mm × 2.1 mm). MS conditions were according to[110]. LC-MS/MS data were acquired and processed using MassLynx 4.2 software (Waters Corp, Manchester, UK) and TargetLynx application (Waters Corp, Manchester, UK). Source data from this study can be accesed at Supplementary Data 2 and at the MetaboIights portal, code MTBLS7533.

**Determination of harmol in tissues.**

a. **Sample treatment**.

For plasma samples, 30 μL were placed in a 1.5 mL polypropylene tube. After that, 20 μL of internal standard (harmine-d₃ prepared in methanol at 1 μg/mL) was added, followed by 250 μL of methanol for protein precipitation. Samples were mixed using a vortex for 60 seconds, and centrifuged at $12,000 \times g$ for 10 min at 4 °C. Finally, supernatants were transferred into glass vials, and 1 μL were injected into the liquid chromatography-tandem mass spectrometry (LC-MS/MS) system.

For tissue samples, frozen tissues were placed in 5 mL glass tubes (200 mg of brain or 100 mg of liver), together with 1 mL of methanol. Samples were homogenized using an Ultra-Turrax dispersing apparatus for 30 s, transferred to 1.5 mL polypropylene tubes, and centrifuged at $12,000 \times g$ for 10 min at 4 °C. Finally, 280 μL of supernatant were transferred into glass vials together with 20 μL of internal standard (harmine-d₃ prepared in methanol at 1 μg/mL), and 1 μL was injected into the LC-MS/MS system.

Matrix-matched calibration curves were prepared in blank extracts in the range of 0.05–250 ng/mL, and injected at least in triplicate.

b. **Instrumentation**.

Extracts were analyzed using an Acquity UPLC I-Class liquid chromatography system (Waters Corp., Milford, MA, USA) interfaced to a Xevo TQ-S Micro (Micromass Waters Corp, Manchester, UK) triple quadrupole mass spectrometer equipped with a Z-Spray electrospray. Chromatographic separation was achieved using an Acquity BEH C18 $2.1 \times 100$ mm, 1.7 μm particle size analytical column (Waters Corp, Wexford, Ireland) maintained at 55 °C. Mobile phases consisted on water (solvent A) and methanol (solvent B), both with 1 mM ammonium formiate and 0.1% formic acid, delivered at a flow rate of 0.3 mL/min, and changing as follows: 0 min 5% B, 0.5 min 5% B, 7.0 min 40% B, 7.1 min 99% B, 8.5 min 99% B, 8.6 min 5% B, and maintained to 10 min for column re-equilibration. Injection volume was 1 μL.

ESI was operated in positive ionization mode using a capillary voltage of 3.0 kV. Nitrogen was used as desolvation (1200 L/h) and cone gas (150 L/h), while source temperature was set to 150 °C, and desolvation temperature to 600 °C. Cone voltage and collision energies, using argon (99.995%) as collision gas, were optimized for each compound. Data were acquired in selected reaction monitoring (SRM) acquisition mode, acquiring at least 3 transitions/compound. Dwell times were selected in order to acquire 12 points/peak, being at least 10 ms/transition. LC-MS/MS data were acquired and processed using MassLynx 4.2 software (Waters Corp, Manchester, UK) and TargetLynx application (Waters Corp, Manchester, UK).

The detailed LC-MS/MS conditions for harmol analysis are detailed in Supplementary Table 2. Source data from this study can be accesed Supplementary Data 3 and at the MetaboLights portal, code MTBLS7533.

**Liver lipid measurement.** 50–100 mg of liver tissue were homogenized in a mix of 200 μl of chloroform and 400 μl of methanol. Then another 200 μl were added to the samples and a new homogenization was performed. 200 μl of water were added, and a third homogenization was performed. Samples were centrifuged during 10 min at $3000 \times g$. After centrifugation, the lower phase was collected and air-dried overnight. The next day, the pellet was weighted in order to determine the total amount of extracted lipids.

For further triglyceride determination, the lipid pellet was dissolved in 250 μl of Triton X-100 (2%) and triglycerides were determined using the BioSystems Triglycerides kit (REF: 11528).

**Histology in kidney paraffin sections.** Kidneys were dissected from mice, fixed in formalin and embedded in paraffin blocks. 5 μm-thick sections were stained with hematoxylin and eosin. Tissue histological characteristics were assessed following Nonneoplasic Lesion Atlas, performed by the National Toxicology Program (https://ntp.niehs.nih.gov/nnl/index.htm). Two representative fields of 20× magnification were analyzed for each slide. Longitudinal planes of kidney were analyzed for the occurrence of immune infiltrates, haemorrhage, and dilatation of tubules or of Bowman capsules. A subjective quantification of the level of damage was assigned to each slide, taking into account the histological findings indicated above. All microscopic evaluations (10–40× magnification) were performed in a blinded manner using the DM2000 LED optical microscope with the DMC5400 Research camera (Leica).

**Histology in muscle cryosections.** Soleus muscle samples were frozen in liquid nitrogen and stored at −80 °C until analysis. Before sectioning, the muscles were incubated in optimal cutting temperature compound (OCT). Transverse serial cross-sections (7 μm thick) were obtained using a cryostat (Microm HM 505 N Cryostat Microtome) maintained at −25 °C and mounted onto glass microscope slides. Cryosections were colored with hematoxylin&eosin solution for measuring cross-sectional area or with NADH stain for fiber typing using established histological techniques.

Morphometry of musculoskeletal fibers was performed on high-resolution micrographic images obtained on a BX50 Olympus Polarized Light Microscope using a Jenoptik Gryphax microscope camera. Images were taken at 4x magnification and microphotograph measurement calibration was performed by means of a stage micrometer diamond ruled with 1 mm divisions. All fibers of the muscle were quantified (an average of 500 fibers per sample) and their cross-sectional area was expressed in μm² using ImageJ 1.53n software. Fiber typing was performed by counting the type I fibers stained with the Bitrotetrazolium Blue chloride in dark blue (representing the NADH activity) and the rest of the fibers corresponding to the type II using ImageJ 1.53 software.

**Statistical analysis**

Statistical analyses were performed using GraphPad Prism 9 for MacOS. Mean differences between two groups were analyzed using two tailed Student's t-test. Differences between three or more groups were analyzed using one-way ANOVA followed by post-hoc analysis through Tukey's multiple comparison correction. Both analyses were performed as paired (repeated measures in the case of ANOVA) or unpaired tests, depending on the type of analyzed samples. Differences between two or more groups along time were calculated using the two-way ANOVA followed by post-hoc analysis through Tukey's multiple comparison correction. Linear correlations were performed using Pearson test.

**Reporting summary**

Further information on research design is available in the Nature Portfolio Reporting Summary linked to this article.

## Data availability

Mass spectrometry data has been deposited at accession code MTBLS7533, and is available in Supplementary Data 2–3. All other data is available in the manuscript, in the Supplementary Figures, and in the Source Data file provided with this paper.

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

## Acknowledgements

We thank Manuel Serrano for his unfaltering support and advice; Fernando Pelaez, Isabel Blanco and Alejo Efeyan for their help in the CNIO facilities; Rosa Serrano and Concepcion Timon for their help with animal procedures; Marta Barradas for her help and support in the laboratory tasks; Laura Formentini and Jose Antonio Enriquez for their kind technical and scientific support; and the *Caenorhabditis* Genetics Center (CGC) at the University of Minnesota for providing *C. elegans* strains. L.F.C.-M. was supported by the IMDEA Food institute and by a PhD Fellowship from the Portuguese Foundation for Science and Technology (FCT-MCTES, SFRH/BD/124022/2016). Work in the laboratory of P.J.F.-M. was funded by the IMDEA Food Institute, the Ramón Areces Foundation (CIVP18A3891), the AECC (SIRTBIO, LABAE18008FERN), the MICINN (SAF2017-85766-R and PRPPID2020-114077RB-I00) co-funded by the European Regional Development Fund, and a Ramon y Cajal Fellowship (MICINN, RYC-2017-22335). J.L.L.-A. was funded by the Spanish Ministry of Science and Innovation (MICINN) (PTA2017-14689-I). Work in Fresh-Age laboratory was funded by: Instituto de Salud Carlos III CB16/10/00435 (CIBERFES), (PID2019-110906RB-I00/AEI/10.13039/501100011033) from the Spanish Ministry of Innovation and Science; FGCSIC/PSLINTERREG/FEDER; PROMETEO/2019/097 de "Consellería de Sanitat de la Generalitat Valenciana" and EU Funded H2020-DIABFRAIL-LATAM (Ref: 825546). Part of the equipment employed in this work has been funded by Generalitat Valenciana and co-financed with ERDF funds (OP ERDF of Comunitat Valenciana). Support from Ramón Areces Fundation and Soria Melguizo Foundation is also acknowledged. E.G.-D. was a recipient of a predoctoral grant financed by the Spanish Ministry (FPU18/05350). Work in the laboratory of IdP was funded by the MICINN (RTI2018-100872-J-I00) and PlaGenT excellence research program of the Valencian regional government (CIDENGENT/2019/044). Work in the laboratory of M.-I.G. was funded by PROMETEU/2018/135 from "Consellería, de Sanitat de la Generalitat Valenciana" and part of the equipment employed in this work has been funded by Generalitat Valenciana and co-financed with ERDF funds (OP ERDF of Comunitat Valenciana 2014–2020). F.M. is grateful to the Austrian Science Fund FWF (SFB-LIPOTOX F3007 & F3012, W1226, P29203, P29262, P27893, and P31727); the Austrian Federal Ministry of Education, Science and Research and the University of Graz for grants Unkonventionelle Forschung-InterFast and ysleep (BMWFW-80.109/0001-WF/V/3b/2015) and the field of excellence program BioHealth. We acknowledge support from NAWI Graz, the BioTechMed-Graz agship project EPIAge. T.E. acknowledges support from Austrian Science Fund FWF (P 33957 and TAI 602 1000). Work in the laboratory of R.H.H. was financially supported by a VIDI grant from ZonMw (no. 91715305). J.S.D. was supported by Ohio University. D.F.-S. acknowledges Ministerio de Universidades in Spain for his Margarita Salas postdoctoral grant (Ref. MGS/2021/15). The CGC group is funded by NIH Office of Research Infrastructure Programs (P40 OD010440).

## Author contributions

L.F.C.-M. designed, performed and analyzed most of the experiments, and worked in the manuscript writing. E.G.-D., J.V. and M.C.G.-C. performed the frailty tests in old mice. RLM and RHH tested lifespan in *C. elegans* worms. J.L.L.-A. helped in performing and analyzing many of the presented experiments. A.B.-G. and IdP determined GABAergic synaptic events in mouse brain slices and performed mouse behavioral analysis testing locomotion and anxiety levels. A.T.-G. and M.-I.G. tested lifespan in D. melanogaster flies. D.F.-S. and OJP measured harmol levels in plasma and tissue samples. T.E., S.J.H., C.M., and F.M. measured polyamines in cells and tissues. J.S.D. constructed the C.

elegans amx triple mutant. J.G., M.P., and D.M. performed and analyzed confocal microscopy experiments. A.P. and A.S.-R. helped in the metabolic characterization of obese mice. D.V.-B. and T.F. helped in the molecular characterization of harmol and its sources. M.I.L. helped in measuring kidney biochemistry. G.H. helped in the PAMPA-BBB assay. P.J.F.-M. designed, coordinated and supervised all experiments, participated in data analysis, gathered funding and wrote the manuscript.

## Competing interests

F.M. is scientific cofounder of Samsara Therapeutics, a company that develops novel pharmacological autophagy inducers. F.M. and T.E. have equity interests in and are advisors of TLL The Longevity Labs GmbH. The remaining authors declare no competing interests.

## Additional information

**Luis Filipe Costa-Machado** [1,2,3], **Esther Garcia-Dominguez** [4], **Rebecca L. McIntyre** [5], **Jose Luis Lopez-Aceituno** [1], **Álvaro Ballesteros-Gonzalez** [6], **Andrea Tapia-Gonzalez** [7], **David Fabregat-Safont** [8,9], **Tobias Eisenberg** [10,11,12], **Jesús Gomez** [13], **Adrian Plaza** [1], **Aranzazu Sierra-Ramirez** [1], **Manuel Perez** [13], **David Villanueva-Bermejo** [14], **Tiziana Fornari** [14], **María Isabel Loza** [2,3], **Gonzalo Herradon** [15], **Sebastian J. Hofer** [10,11,12], **Christoph Magnes** [16], **Frank Madeo** [10,11,12], **Janet S. Duerr** [17], **Oscar J. Pozo** [8], **Maximo-Ibo Galindo** [6,18,19], **Isabel del Pino** [7,20], **Riekelt H. Houtkooper** [5], **Diego Megias** [13], **Jose Viña** [4], **Mari Carmen Gomez-Cabrera** [4] & **Pablo J. Fernandez-Marcos** [1] ✉

[1]Metabolic Syndrome Group – BIOPROMET. Madrid Institute for Advanced Studies - IMDEA Food, CEI UAM + CSIC, E28049 Madrid, Spain. [2]Kaertor Foundation, EMPRENDIA Building, Floor 2, Office 4, Campus Vida, E-15706, Santiago de Compostela, Spain, E-15706 Santiago de Compostela, Spain. [3]BioFarma Research Group, Center for Research in Molecular Medicine and Chronic Diseases (CIMUS), Universidade de Santiago de Compostela, Santiago de Compostela, Spain. [4]Freshage Research Group, Department of Physiology, Faculty of Medicine, CIBERFES, Fundación Investigación Hospital Clínico Universitario/INCLIVA, University of Valencia, Valencia, Spain. [5]Laboratory Genetic Metabolic Diseases, Amsterdam Gastroenterology, Endocrinology, Metabolism, Amsterdam Cardiovascular Sciences, Amsterdam UMC, University of Amsterdam, Amsterdam, Netherlands. [6]Developmental Biology and Disease Models Group, Centro de Investigación Príncipe Felipe, 46012 Valencia, Spain. [7]Neural Plasticity Group, Centro de Investigación Príncipe Felipe, 46012 Valencia, Spain. [8]Applied Metabolomics Research Group, Hospital del Mar Medical Research Institute - (IMIM), Barcelona, Spain. [9]Environmental and Public Health Analytical Chemistry, Research Institute for Pesticides and Water, University Jaume I, 12006 Castelló de la Plana, Castellón, Spain. [10]Institute of Molecular Biosciences, NAWI Graz, University of Graz, Humboldtstraße 50, 8010 Graz, Austria. [11]BioTechMed Graz, 8010 Graz, Austria. [12]Field of Excellence BioHealth – University of Graz, Graz, Austria. [13]Confocal Microscopy Unit, Biotechnology Programme, Spanish National Cancer Research Centre (CNIO), Melchor Fernández Almagro 3, 28029 Madrid, Spain. [14]Department of Production and Characterization of Novel Foods, Institute of Food Science Research (CIAL UAM-CSIC), C/ Nicolás Cabrera, 9, P.O. Box. 28049 Madrid, Spain. [15]Lab. Pharmacology, Faculty of Pharmacy, Universidad CEU San Pablo, Urb. Montepríncipe, 28668 Boadilla del Monte, Madrid, Spain. [16]HEALTH-Institute for Biomedicine and Health Sciences, Joanneum Research Forschungsgesellschaft mbH, 8010 Graz, Austria. [17]Department of Biological Sciences, Ohio University, Athens, OH 45701, USA. [18]Instituto Interuniversitario de Investigación de Reconocimiento Molecular y Desarrollo Tecnológico (IDM), Universitat Politècnica de València, Universitat de València, 46022 Valencia, Spain. [19]UPV-CIPF Joint Research Unit "Disease Mechanisms and Nanomedicine". Centro de Investigación Príncipe Felipe, 46012 Valencia, Spain. [20]Instituto de Neurociencias, Consejo Superior de Investigaciones Científicas, Universidad Miguel Hernández, Campus de Sant Joan, 03550 Alicante, Spain. ✉e-mail: pablojose.fernandez@imdea.org

