## [Peer Review File · Nature Communications]

Peripheral modulation of antidepressant targets MAO-B and GABAAR by harmol induces mitohormesis and delays aging in preclinical modelsREVIEWER COMMENTS

Reviewer #1 (Remarks to the Author):

In this manuscript, the authors used C2C12 cells loaded with a carefully established concentration of TMRM to screen a library of 982 compounds in triplicate, with FCCP as a positive control, for transient depolarization, to identify 36 compounds also matching the latter criteria (i.e. reversibility). From these and followed by the Seahorse mitostress protocol, they identified an unnamed number of compounds, including several beta-carbolines, amongst these harmol, which they further characterize in more detail.

They study several mitohormesis-related response pathways, including mitophagy and the canonical AMPK pathway, paralleled by induction of TFAM, as well as increases in components of complex II and V, the latter likely linked to increased ATP synthesis, as experimentally shown.

Inspired by the known role of harmine in inhibiting MAO, they find (solely based on inhibitors, but not shRNAs nor Crispr/CAS9 lines) MAO(-B) and potentially GABA-R to execute the effects of harmol.

They then show favourable effects of harmol in regards to improvement of lipid and glucose metabolism in HFD mice, which are compared to vehicle- and rosiglitazone-treated animals.

Known inducers of mitohormesis (e.g. metformin) have not been tested, but to requests this now would certainly be beyond the scope of the manuscript.

Lastly, they show harmol to extend lifespan in *C. elegans* and *Drosophila*, as well as remarkable effects of harmol on metabolism and fitness in aged mice.

This manuscript contains very interesting findings, is based on an extensive body of work, and especially the use of three different model organisms convincingly establishes the lead compound, harmol, as a promising agent to delay aging per se, as well as aging-associated mortalities.

My key concern is the questionable specificity of the (already combined) MAO/GABAR pathway proposed.

I would suggest the following experiments:

Given the unclear mechanistic contribution of AMPK:

Does harmol extend lifespan in aak-2/AMPK-deficient nematodes?

Optional: Does harmol induce aak-2 phosphorylation? The CellSignaling phospho-AMPK-AB works nicely in nematodes, whereas the basal does not. The authors may use tubulin instead, and prove specificity by including a aak-2-derived worm pellet.

Role of MAO:

Nematodal AMX occurs in three isoforms, and is considered the *C. elegans* orthologue of MAOs.

There (luckily) is a triple mutant for all three isoforms available:

<https://wormbase.org/resources/paper/WBPaper00030371#0--10>

Does application of harmol or harmine to amx(-triple)-mutants of *C. elegans* still extend lifespan? Or is it reduced? If so, to which extent (50th and 90th percentile)?

Conversely, does AICAR extend lifespan in amx(-triple)-mutants (compared to N2)? Details on AICAR

concentrations in *C. elegans* can be found here:

<https://journals.plos.org/plosone/article?id=10.1371/journal.pone.0148089>

Optional: Does the MAO-B inhibitor rasagilin extend lifespan in *C. elegans* N2? I.a.w., why was rasagilin replaced by selegilin in the experiments depicted?

Optional: Does the pan-MAO inhibitor harmine extend lifespan in *C. elegans* N2?

Role of GABA-R:

UNC-49 is the *C. elegans* orthologue of the mammalian GABA-receptor:

<https://wormbase.org/search/all/unc-49>

Does application of harmol to unc-49 mutants still extend *C. elegans* lifespan? Or is it reduced? If so, to which extent (50th and 90th percentile)?

Conversely, does AICAR extend lifespan in unc-49 mutants (compared to N2)?

Optional: Does the GABAR inhibitor gabazine extend lifespan in *C. elegans* N2? I.a.w., why was gabazine replaced by FG-7142 in the experiments depicted?

Optional: Experiments in *Drosophila* should be considered by analogy; however and to my understanding, there is no MAO orthologue known in flies (which puts the proposed mechanism into question, though from a different angle; this should at least be discussed).

Optional but important if samples are still available: Is mitochondrial mass (as by mtDNA/nDNA) increased in samples from Figs. 6a, 6b, and 6e-n (combined & post only)?

Further comments:

Please provide a concise list of all compounds (incl. names and CAS numbers) that were a) initially used, and identified by b) the primary, as well as c) the mitostress assay.

Screen: Why were 1.3 ug/ml used, instead of a specific molarity - apparently the purity and formulas were available?

Please provide the rationale based on which the beta-carbolines were preferred over other candidates. Also, how do AICAR, harmine and gabazine perform in these assays (Fig. 3h and i show only the 60 min affect on TMRM)? I.a.w. would they have been identified if contained in the library?

Suggested text edits:

Line 90 - Reference [4] should be replaced by PubmedID 17908557 as the primary reference.

Line 104 - Reference [17] should be replaced by PubmedID 19433800 as the primary reference.

Lines 99 - 104: additional prime examples of mitohormesis are the green tea polyphenols ECG and EGCG (PubMedID 28992589) as well as D-glucosamine (PubMedID 24714520).

Discussion: The beta-carboline harmine has been shown to extend lifespan by ameliorating bacterial infection & could be mentioned in the discussion (PubMedID 23544153).

Reviewer #2 (Remarks to the Author):

This is a potentially interesting study which attempts to establish a link between metabolism, psychiatric status and longevity. In this work, authors found Harmol was a bioactive compound to induce transient mitochondrial depolarization and has some effects in animal experiments. The study is, however, descriptive in nature and lacks sufficient novelty and mechanistic insights which may be more suitable for specialized journals. Still here are some questions need to be addressed:

- 1) What are the phenotypes of knock out or knock down animals for MAO-B and/or GABAAR? Do the phenotypes corroborate the authors' hypothesis?
- 2) Why does inhibition to both MAO-B and GABAAR required for the observed effects?
- 3) Low concentration of Harmol may bind to its high affinity target such as DYRK1A. How do the authors rule out the possible involvement and action through DYRK1A?
- 4) Does Rosig regulate mitochondrial depolarization in the screening assay? What is the rationale to use Rosig as the reference compound ?
- 5) What were the concentrations of Harmol in blood and CSF when 100mg/Kg dose was used?

Reviewer #3 (Remarks to the Author):

The manuscript by Costa-Machado et al. provides acceptable evidence that Harmol, a member of the beta-carbolines family with anti-depressant properties, improves mitochondrial function and metabolic parameters and extends healthspan. Yet, I find that the authors can improve the manuscript on the following basis

Figure 1: The author relies too heavily on TMRM, measured at 16 hours, to indicate mitochondrial function. Please, use an alternative assay such as JC1 to confirm high-throughput screen is specific to the mitochondrial program.

Figure 2: Autophagy is indeed modulated by PINK1. But there exists an entire array of molecules, e.g., parkin, which determine autophagic flux mentioned in the manuscript. In the discussion section, please provide additional support for autophagy claims or expand your thoughts on autophagy upon treatment with harmol.

Figure 2: The author asserts that increases and decreases in ATP content were due to increase mitochondrial content but does not provide molecular support. After treatment, please use electron transport chain inhibitors to verify ATP production via mitochondria instead of other sources, e.g., anaerobic respiration. Alternative measure lactate content, which should decrease upon increased production of ATP to signify upregulation of mitochondria-ATP.

What is the effect of harmol on differentiated myotube? Do myotubes increase or decrease in size? It is essential to clarify that observed differences are not due to myotube-hypertrophy.

Figure 4 and corresponding results section: Please mention mice information, add references and

explain why the authors have chosen 3 hours and 7-hour duration?

Figure 5 and corresponding results section: Please be consistent about the treatment duration in the texts – 8 weeks or 16 weeks? Authors have performed necropsy or noticed any marked differences at the endpoint as the aged C57BL/6J mice are prone to cancer and other morbidities – also, were there any mortalities? Decrease in fluid intake is a primary reason for decreased weights? Please provide an H&E stain for kidneys to confirm that harmol does not damage kidneys. If possible, show blood chemistry to establish homeostasis.

Figure 5: In vitro experiments showed significant effects of harmol on myotubes but in vivo experiments failed to consider any skeletal muscle protein. Please examine the impacts of harmol aged skeletal muscle translation—for example, measures of myogenin, actinin, or MyHC. Although metabolic panels were provided, this could be misleading because the cause and effects are primarily hidden in whole-tissue assays. Furthermore, what is the impact of harmol on muscle fiber typing (fast and slow-twitch, muscle cross-sectional area)? Switch of fast to slow-twitch or vice versa may affect metabolism measures. Please examine muscle fibers type in-depth upon treatment with harmol.

Figure 6: Please explain how the authors have determined the samples size for the behavioral tests? I believe fewer samples in each sex have made the authors combine the male and female mice for the behavior data? Explain why Tukey correction was done for rotarod and grip strength and not for ladder climbing, grid hanging, and running tests? It is exciting to see the marked difference in grid hanging duration after four months of harmol treatment, but at the same time, it brings the question of how could these 27-month-old (~45 gram) mice hang for more than 5 minutes?

Remarkable life-extending effects observed in *C. Elegans* worms and *D. melanogaster*. Do you see the effects in telomerase expression or activity?

Does harmol act directly or break down into metabolites activating observed signaling. Please clarify.

Does harmol affect amino acids as it did on glucose and lipids? Amino acids connect metabolism to neuro-activation. Example tryptophan, aspartate, & glutamate have substantial effects on the GABA, AMPK, LC3. Please state your opinions on these straightforward possibilities.

Please review entire documents for typos, run-ons, and repetitions. In addition, please use punctuation and avoid complex sentences.

Reviewer #4 (Remarks to the Author):

In this study, authors found harmol, a member of the beta-carbolines family with anti-depressant

properties, as a novel mitohormetic compound. Harmol induced a transient mitochondrial depolarization, mitophagy and AMPK activation in cultured C2C12 myotubes, in the mouse liver, BAT and muscle. The effects of harmol treatment on mitochondria were mimicked by simultaneous modulation of monoamine-oxidase B and GABAAR receptor. Harmol treatment improved symptoms in diet-induced prediabetic mice. Harmol or a combination of MAO-B and GABAAR modulators extended the lifespan of *Caenorhabditis elegans* or *Drosophila melanogaster*, and delayed frailty onset in mice. They concluded that these results reveal a link between improved psychiatric status and healthspan extension through peripheral mitohormesis.

The effects of harmol are novel and provide mechanistic significant insights into mitohormesis. Their results support the interaction between Harmol and MAO-B and GABAAR to extend healthspan via its mitohormetic action in the peripheral tissues. However, it was not clear how this study revealed a link between psychiatric status and healthspan extension through peripheral mitohormesis (lines 68-70). Although the mitohormetic effect was shown in the peripheral tissues, the effect of harmol treatment on mitochondria in the nervous system is not investigated. Also, how harmol affects on mouse behavior related to anxiety are not shown. Thus, this work does not seem to reveal a link between psychiatric status and peripheral mitohormesis. Detailed comments are listed below.

Major points

1. Authors showed harmol passed BBB to some extent ($\text{LogPe} > -4.7$, $\text{Ca(t)/Cd(0)} * 100 = 24.64\%$), or it also affects mitochondrial homeostasis in the nervous system?
2. Is there any correlation between mitophagy in the peripheral tissues and psychiatric status after treatment with Harmol?
3. Figure 4. Harmol and Rosiglitazone show different effects on PINK1 levels in the gastrocnemius: is it expected? Please discuss.
4. Figure 6. Control for Harmol was DMSO in a, b, c, and in Figure 6d, S6d,f, and Table S2d,e: control treatment was with EtOH. Figures S6d and 6c, EtOH changed to DMSO for WT *D. melanogaster*, information on the figure is not consistent, while data is from the same experiment.
5. Lines 555-557: It seems to be a wrong statement. According to data in table S1 table, harmine is totally impermeable $\text{logPe} < -4.7$, but harmol is not. So, it is not clear how it is affected CNS.

Minor comments:

1. Line 352: Possibly “-” was missed “($\text{LogPe} < -4.7$)”, because it was stated that $\text{LogPe} < -4.7$ refers to a totally impermeable compound
2. Line 458: S1a -> S2a; Line 462: S1b -> S2b and further, S1 table was BBB permeability table.
3. Misspellings: Line 697: carried our -> carried out, line 196: ith -> with.

POINT-BY-POINT RESPONSE TO REVIEWER COMMENTS

We thank the reviewers for their insightful comments. We consider they are very well taken and add more depth to our findings. We also thank the reviewers for their positive comments regarding our manuscript: *“This manuscript contains very interesting findings, is based on an extensive body of work, and especially the use of three different model organisms convincingly establishes the lead compound, harmol, as a promising agent to delay aging per se, as well as aging-associated mortalities”* (Rev. #1); *“This is a potentially interesting study”* (Rev. #2); or *“The manuscript by Costa-Machado et al. provides acceptable evidence that Harmol, a member of the beta-carbolines family with anti-depressant properties, improves mitochondrial function and metabolic parameters and extends healthspan”* (Rev. #3).

Reviewer #1:

[REVIEWER]: Known inducers of mitohormesis (e.g. metformin) have not been tested, but to requests this now would certainly be beyond the scope of the manuscript.

[AUTHORS]: We agree with this reviewer that other mitohormetic compounds, such as metformin, could have served as additional controls. In this regard, we have used rosiglitazone, a well-described mitohormetic compound¹ with anti-diabetic potential^{2,3}.

[REVIEWER]: Lastly, they show harmol to extend lifespan in *C. elegans* and *Drosophila*, as well as remarkable effects of harmol on metabolism and fitness in aged mice.

This manuscript contains very interesting findings, is based on an extensive body of work, and especially the use of three different model organisms convincingly establishes the lead compound, harmol, as a promising agent to delay aging per se, as well as aging-associated mortalities.

[AUTHORS]: We heartfully thank this reviewer for these positive comments on our work.

[REVIEWER]: My key concern is the questionable specificity of the (already combined) MAO/GABAR pathway proposed.

I would suggest the following experiments:

Given the unclear mechanistic contribution of AMPK:

Does harmol extend lifespan in *aak-2*/AMPK-deficient nematodes?

Optional: Does harmol induce *aak-2* phosphorylation? The CellSignaling phopho-AMPK-AB works nicely in nematodes, whereas the basal does not. The authors may use tubulin instead, and prove specificity by including a *aak-2*-derived worm pellet.

[AUTHORS]: Following these suggestions, we first tested lifespan in *aak-2*-deficient nematodes⁴ treated with vehicle (0.1% DMSO) or with 15 $\mu\text{g/ml}$ harmol. As shown in new Figure 6c, S6g and Table S3c, ablation of *aak-2* prevented harmol-mediated lifespan extension, indicating that AMPK/AAK-2 activity is essential for harmol effects.

In addition, we tested the effects of harmol treatment on AAK-2 activation by Western blot. As shown in new Figure S6b, worms treated with harmol for 1 week showed a slightly but not significant reduction in AAK-2 phosphorylation. Since we have shown that long-term treatment with harmol increased ATP production and mitochondrial function (Figure 1d and 2h), we interpret this slight decrease in AAK-2 phosphorylation as a consequence of an improvement in mitochondrial function by

long-term treatment with harmol. We discuss about these findings in the Discussion section.

[REVIEWER]: Role of MAO:

Nematodal AMX occurs in three isoforms, and is considered the *C. elegans* orthologue of MAOs.

There (luckily) is a triple mutant for all three isoforms available:

<https://wormbase.org/resources/paper/WBPaper00030371#0--10>

Does application of harmol or harmine to amx(-triple)-mutants of *C. elegans* still extend lifespan? Or is it reduced? If so, to which extent (50th and 90th percentile)?

[AUTHORS]: Following this suggestion, we have established a collaboration with Prof. Duerr, who has generated this triple mutant of the three homologues of MAO in *C. elegans* and kindly donated them for our experiments. We observed that ablation of the three MAO homologues in worms did not prevent harmol-mediated lifespan extension (Figure 6e, S6i and Table S3e). These results are in line with harmol function as a MAO inhibitor: absence of MAO mimics harmol, and does not affect its function.

[REVIEWER]: Conversely, does AICAR extend lifespan in amx(-triple)-mutants (compared to N2)? Details on AICAR concentrations in *C. elegans* can be found here:

<https://journals.plos.org/plosone/article?id=10.1371/journal.pone.0148089>

Optional: Does the MAO-B inhibitor rasagilin extend lifespan in *C. elegans* N2? I.a.w., why was rasagilin replaced by selegilin in the experiments depicted?

Optional: Does the pan-MAO inhibitor harmine extend lifespan in *C. elegans* N2?

[AUTHORS]: To answer these requests, we treated *amx-1/2/3* mutant worms with 1mM AICAR and found no increase in lifespan, as shown in Figure S6i and Table S3e. This result fits with our hypothesis of harmol being an inhibitor of monoamine oxidase activity, and thus loss of MAO activity does not affect harmol effects on lifespan.

About our reasons to use selegilin in Figure 6: we decided to use selegilin when planning experiments with the MAO inhibitors in *C. elegans* lifespan extension, because selegilin has already proved to extend lifespan in previous reports⁵, while there were no previous lifespan experiments with rasagilin. This way, we could contrast our results with previous bibliography.

[REVIEWER]: Role of GABA-R:

UNC-49 is the *C. elegans* orthologue of the mammalian GABA-receptor:

<https://wormbase.org/search/all/unc-49>

Does application of harmol to unc-49 mutants still extend *C. elegans* lifespan? Or is it reduced? If so, to which extent (50th and 90th percentile)?

Conversely, does AICAR extend lifespan in unc-49 mutants (compared to N2)?

Optional: Does the GABA-R inhibitor gabazine extend lifespan in *C. elegans* N2? I.a.w., why was gabazine replaced by FG-7142 in the experiments depicted?

[AUTHORS]: Indeed, when *unc-49* mutant worms are treated with harmol, they did not extend their lifespan (Figure 6d, S6h and Table S3d). These results are in agreement with the blunted harmol-induced mitochondrial depolarization after treatment with the GABAAR inhibitor bicuculine, as shown in Figure 3f-g, and support the importance of the GABAAR/UNC-49 activity in harmol-mediated lifespan extension. This also answers this reviewer's comment on the use of gabazine and FG-7142: gabazine is a GABAAR

inhibitor, that blocks the ion channel. In contrast, FG-7142 is an inverse agonist of GABAAR: it induces an inverse flow of ions through the channel, that must remain functional for FG-7142 to work. Our hypothesis is that harmol is, as FG-7142, a GABAAR inverse agonist, and these experiments with *unc-49* mutants support this hypothesis: a dysfunctional GABAAR/UNC-49 channel impairs harmol effects, as happened in our bicuculin experiments.

[REVIEWER]: Optional: Experiments in *Drosophila* should be considered by analogy; however and to my understanding, there is no MAO orthologue known in flies (which puts the proposed mechanism into question, though from a different angle; this should at least be discussed).

[AUTHORS]: As this reviewer accurately points out, there is no straight homologue of mammalian MAO in *Drosophila*⁶. However, several enzymatic activities substitute for MAO in the fly⁷⁻⁹, and MAO inhibitors in mammals have been shown to work also in flies^{10,11}. We therefore hypothesize that the MAO inhibitors, including harmol, used in our work act on the MAO-like activities in our *Drosophila* models and have an equivalent impact in lifespan extension. We have included this commentary in our new manuscript, lines 755-760.

[REVIEWER]: Optional but important if samples are still available: Is mitochondrial mass (as by mtDNA/nDNA) increased in samples from Figs. 6a, 6b, and 6e-n (combined & post only)?

[AUTHORS]: To answer this question, we treated flies and worms with harmol for 1 or 2 weeks, respectively, and extracted their total DNA to measure the ratio between mitochondrial and nuclear DNA. As shown in new Figure S6d, f, this ratio was not altered after harmol treatment in any of the two species, in the same line as observed with C2C12 myotubes (Figure 2g) and with mouse tissues (Figure 4j). We have included this new data in Figure S6d, f, and commented on them in the main manuscript discussion section.

[REVIEWER]: Further comments:

Please provide a concise list of all compounds (incl. names and CAS numbers) that were a) initially used, and identified by b) the primary, as well as c) the mitostress assay.

[AUTHORS]: We have included a new table (Supplemental Table S4) with all the compounds included in the commercial libraries and used for our screening. Due to intellectual property protection, we cannot disclose all the compounds that were identified in the subsequent stages of our screening, other than the ones shown in the article in Figure 1 and S1.

[REVIEWER]: Screen: Why were 1.3 ug/ml used, instead of a specific molarity - apparently the purity and formulas were available?

[AUTHORS]: Libraries were purchased in mg/mL concentrations, and a common concentration in mass was therefore used for the screening. Further experiments kept the original effective concentration (1.3 µg/mL).

[REVIEWER]: Please provide the rationale based on which the beta-carbolines were preferred over other candidates.

[AUTHORS]: Harmol was the most robust compound in our Seahorse experiments, with very reproducible increase in cellular spare respiratory capacity. Also, there were three members of the same family identified in the screening, strengthening the structural relevance of this compound. We have justified this choice in the main text when describing our screening results.

[REVIEWER]: Also, how do AICAR, harmine and gabazine perform in these assays (Fig. 3h and i show only the 60 min affect on TMRM)? I.a.w. would they have been identified if contained in the library?

[AUTHORS]: To answer this reviewer's question, we have tested the effects the indicated compounds on TMRM signal in differentiated C2C12 cells.

First, we compared harmol and harmine. As shown in Figure to reviewers 1a, harmine depolarized mitochondria, although this effect took longer than with harmol. We tested rosiglitazone effects in separate, since this compound is the positive control for many following experiments. As expected, we saw a clear mitochondrial depolarization ability, as shown in new Figure S1j.

Finally, we tested AICAR (an AMPK activator) and gabazine (a GABAAR inhibitor), and did not find any mitochondrial depolarization (Figure to reviewers 1b). These effects were expected: AMPK is activated upon mitochondrial depolarization; but AMPK activation by AICAR does not need any mitochondrial depolarization to take place. In turn, inhibition of the GABAAR is not necessarily linked to mitochondrial depolarization, as shown in this experiment.

Figure 1 for reviewers. Effect of requested compounds on TMRM mitochondrial polarization. (a, b) Differentiated C2C12 myotubes were stained with TMRM for 16 hours, treated with the indicated compounds, and TMR fluorescence was measured at the indicated times after compound addition. Line-connected dots represent the average of the indicated number of individual replicates. Error bars represent the standard deviation.

Suggested text edits:

[REVIEWER]: Line 90 - Reference [4] should be replaced by PubmedID 17908557 as the primary reference.

[AUTHORS]: We have added this reference (new reference [5]).

[REVIEWER]: Line 104 - Reference [17] should be replaced by PubmedID 19433800 as the primary reference.

[AUTHORS]: We have added this reference (new reference [21]).

[REVIEWER]: Lines 99 - 104: additional prime examples of mitohormesis are the green tea polyphenols ECG and EGCG (PubMedID 28992589) as well as D-glucosamine (PubMedID 24714520).

[AUTHORS]: For ECG and EGCG we have chosen a paper dealing with both compounds: PMID 34607977 (new reference [19]). We have also included the suggested reference for D-glucosamine supplementation as reference [20].

[REVIEWER]: Discussion: The beta-carboline harmine has been shown to extend lifespan by ameliorating bacterial infection & could be mentioned in the discussion (PubMedID 23544153).

[AUTHORS]: We have commented this finding in our discussion section, and included this new reference [81].

Reviewer #2 (Remarks to the Author):

This is a potentially interesting study which attempts to establish a link between metabolism, psychiatric status and longevity. In this work, authors found Harmol was a bioactive compound to induce transient mitochondrial depolarization and has some effects in animal experiments. The study is, however, descriptive in nature and lacks sufficient novelty and mechanistic insights which may be more suitable for specialized journals. Still here are some questions need to be addressed:

[REVIEWER]: 1) What are the phenotypes of knock out or knock down animals for MAO-B and/or GABAAR? Do the phenotypes corroborate the authors' hypothesis?

[AUTHORS]: We have tested harmol effects on lifespan in the MAO homologue mutants *amx-1/2/3* and in the GABAAR homologue *unc-49* mutant worms. As shown in new Figure 6e, S6h and Table S3d, ablation of the GABAAR completely prevented harmol-mediated lifespan extension. This result is in line with the function of harmol as an inverse agonist of GABAAR, and with the need of a functional GABAAR to achieve harmol effects shown in Figure 3e-g. In contrast, ablation of the three homologues of MAO (*amx-1/2/3*) in *C. elegans* did not prevent harmol-mediated lifespan extension (new Figure 6d, S6i and Table S3e), in line with the function of harmol as a MAO inhibitor (Figure 3a-b).

[REVIEWER]: 2) Why does inhibition to both MAO-B and GABAAR required for the observed effects?

[AUTHORS]: Our results show that single inhibition of MAO (using MAO inhibitors as in Figure 3a-b or MAO mutants as in Figure 6e); or single treatment with the GABAAR inverse agonist FG-7142, as in Figure S3e, did not reproduce harmol effects. We need to combine these two effects to exert mitochondrial depolarization, as shown in Figure 3i-j. We do not know yet the exact reasons for the need of this simultaneous inhibition. We hypothesize that harmol disrupts mitochondrial potential by altering the ion flux through the GABAAR. At the same time, harmol induces an accumulation of polyamines specifically at the mitochondria due to the inhibition of the outer membrane-residing, polyamine-metabolizing MAO activity. It is only when these two simultaneous effects take place that harmol triggers mitohormesis and all the beneficial effects shown in our work.

[REVIEWER]: 3) Low concentration of Harmol may bind to its high affinity target such as DYRK1A. How do the authors rule out the possible involvement and action through DYRK1A?

[AUTHORS] This is a very good point. Indeed the β -carboline harmine is a well-known inhibitor of DYRK1A^{12,13}. In a recent screening searching for harmine analogs with ability to inhibit DYRK1A in neuroblastoma and glioblastoma cells, harmol was identified as an inhibitor of DYRK1A with low effectiveness against the monoamine oxidase MAO-A (but its effects on the MAO-B isoform was not tested)¹⁴. To check if harmol effects on mitochondrial depolarization could be caused by DYRK1A inhibition, we treated C2C12 cells with harmol for 5', 10' and 15', and checked DYRK1A phosphorylation levels. As shown in Figure S3a, we did not detect any significant decrease in DYRK1A phosphorylation. These results indicate that inhibition of DYRK1A was not necessary for

the mitochondrial depolarization induced by harmol in C2C12 myotubes. We have discussed these results in the Discussion section, lines 290-300.

[REVIEWER]: 4) Does Rosig regulate mitochondrial depolarization in the screening assay? What is the rationale to use Rosig as the reference compound?

[AUTHORS] Rosiglitazone is a mitohormetic compound, able to induce mitochondrial biogenesis and improve mitochondrial function^{15,16}. It also proved to behave as a potent and reproducible positive control in our cell culture experiments, including mitochondrial depolarization by TMRM (new Figure S1j), increased spare respiratory capacity (Figure S1k); and fulfilled all the *in vitro* mitohormetic biomarkers in Figure 2. Therefore, we consider that rosiglitazone is a good positive control for our mitohormetic effects.

[REVIEWER]: 5) What were the concentrations of Harmol in blood and CSF when 100mg/Kg dose was used?

[AUTHORS] To answer this relevant question, we decided to measure the levels of harmol in different tissues from the mice shown in Figure 4: plasma, liver and brain. We tried to measure harmol in CSF, but the amount and purity of our samples were too low, and we decided to study harmol levels in whole brain and assume that it reflected the brain availability with sufficient fidelity. As shown in Figure 4a-b, harmol showed a poor brain-blood AUC ratio (1.125) in the time-course experiment from 3 to 24 hours after oral administration of harmol, in accordance with the results obtained in the *in vitro* PAMPA-BBB assay shown in Table S1.

Reviewer #3 (Remarks to the Author):

The manuscript by Costa-Machado et al. provides acceptable evidence that Harmol, a member of the beta-carbolines family with anti-depressant properties, improves mitochondrial function and metabolic parameters and extends healthspan. Yet, I find that the authors can improve the manuscript on the following basis

[REVIEWER]: Figure 1: The author relies too heavily on TMRM, measured at 16 hours, to indicate mitochondrial function. Please, use an alternative assay such as JC1 to confirm high-throughput screen is specific to the mitochondrial program.

[AUTHORS]: We have used TMRM as a first screening step to identify compounds reversibly depolarizing mitochondria at short times (30-60 minutes), and allowing for mitochondria recovery at longer times (16 hours). To our knowledge, TMRM is the less aggressive dye to track polarized mitochondria with time; while other dyes, as JC-1 or MitoTracker, are more damaging to live mitochondria¹⁷. According to this same revision, *“typical JC-1 usage as a ratiometric probe for $\Delta\psi_m$ relies on aggregation (threshold) effects, which may render it ill-suited to making reliable distinctions between subtle gradations in $\Delta\psi_m$. It also means that in theory, and as shown in practice¹⁸ and from our own unpublished data, getting the probe to properly work in this fashion is highly sensitive to probe loading concentrations and loading times”. [...] “For these reasons, and assuming adequate equilibration time is allowed, an investigator may find that JC-1 is best suited to more coarse “yes/no” assessments with the goal of determining whether mitochondria populations are either largely polarized or largely unpolarized (e.g., apoptosis-type studies), rather than discriminating finer gradations or differences in membrane potential across regions or populations, for which TMRM is likely better suited”*. Since we are looking for mild, reversible mitochondrial depolarization, we have chosen TMRM as the best option.

Moreover, at 16 hours after treatment we did not use only TMRM. We also used Seahorse, another functional assay to measure mitochondrial fitness, as shown in Figure 1d; and measured by Western blot markers of mitochondrial fitness, as TFAM and complex subunits, as shown in Figure 2f. We consider that the combination of TMRM at short and long times, Western blot and Seahorse is a very robust method to measure mitochondrial function, as proven further by the consistent and clear effects in physiological tests in several animal species.

[REVIEWER]: Figure 2: Autophagy is indeed modulated by PINK1. But there exists an entire array of molecules, e.g., parkin, which determine autophagic flux mentioned in the manuscript. In the discussion section, please provide additional support for autophagy claims or expand your thoughts on autophagy upon treatment with harmol.

[AUTHORS] In this regard, we have used several markers of autophagy to monitor how harmol regulated this process: LC3 lipidation, PINK-1 levels and lysosome-mitochondria co-localization. However, as pointed out by this reviewer, mitophagy is an extremely complex process, that involves a large range of proteins as Parkin, OPA-1 and many others. A deeper, more thorough study of how harmol affects these pathways would indeed be very valuable. We have included these comments in the Discussion section.

[REVIEWER]: Figure 2: The author asserts that increases and decreases in ATP content were due to increase mitochondrial content but does not provide molecular support. After treatment, please use electron transport chain inhibitors to verify ATP production via mitochondria instead of other sources, e.g., anaerobic respiration. Alternative measure lactate content, which should decrease upon increased production of ATP to signify upregulation of mitochondria-ATP.

[AUTHORS] Our results indicate that harmol effects are independent of mitochondrial content, that is not altered by harmol treatment in differentiated C2C12 myotubes (Figure 2g) or in liver, skeletal muscle or BAT (Figure S4g). To test if harmol effects required functional mitochondrial ETC to take place, following this reviewer's suggestion, we treated C2C12 myotubes with DMSO, harmol or harmol combined with the ETC inhibitors rotenone+antimycin for 1 hour. As shown in new Figure S3h, harmol treatment increased ATP production of cells, as already shown; and the addition of the ETC inhibitors completely blunted this effect. These results indicate that a functional ETC is necessary for harmol effects on mitochondrial function.

[REVIEWER]: What is the effect of harmol on differentiated myotube? Do myotubes increase or decrease in size? It is essential to clarify that observed differences are not due to myotube-hypertrophy.

[AUTHORS] This is a good point. To answer it, we treated our C2C12 differentiated myotubes with harmol and took photographs at different times. Then, we measured the size of the myotubes to determine if harmol treatment was affecting it. As shown in new Figure S2f and described in the main text, harmol treatment did not alter myotube size up to 24 hours after treatment.

[REVIEWER]: Figure 4 and corresponding results section: Please mention mice information, add references and explain why the authors have chosen 3 hours and 7-hour duration?

[AUTHORS] We apologize for the missing mouse information in this figure. We have included the number of mice used for each treatment in the figure panels and in the legend. We have included precise information about the mice used in this assay in the text, as well as an explanation of why we chose these time points. Based on the biodistribution profile shown in Figure 4a-b, we consider that these time points are informative of the metabolism of harmol.

[REVIEWER]: Figure 5 and corresponding results section: Please be consistent about the treatment duration in the texts – 8 weeks or 16 weeks?

[AUTHORS]: We have indicated in the text and the figure legend that the treatment with harmol and rosiglitazone was 3 months. We hope this is clear now.

[REVIEWER]: Authors have performed necropsy or noticed any marked differences at the endpoint as the aged C57BL/6J mice are prone to cancer and other morbidities – also, were there any mortalities? Decrease in fluid intake is a primary reason for decreased weights? Please provide an H&E stain for kidneys to confirm that harmol does not damage kidneys. If possible, show blood chemistry to establish homeostasis.

[AUTHORS]: We did not detect any morbidity or mortality in mice treated with harmol, rosiglitazone or vehicle during the length of our experiment, as indicated in the main

text, lines 470-472. During necropsy, we did not find any macroscopic evidence of cancer either.

We observed a slight decrease in fluid intake at the beginning of the treatment, that reversed after a few weeks, as shown in Figure S5a. This decrease was not accompanied by altered food intake (Figure S5b). Therefore, we do not think that the reduced body weight observed in harmol-treated mice was due to decreased fluid or food intake. Rather, we propose that a more efficient energy management, reflected in decreased energy expenditure (Figure 5j) and improved glucose homeostasis (Figure 5e-i) is the main cause of the decreased body weight of harmol-treated mice.

Finally, to answer this reviewer's question about kidney structure and function, we stained kidney sections with hematoxylin & eosin, and analyzed different histological findings: immune infiltrates, haemorrhage, and dilatation of tubules or of Bowman capsules. As shown in Figure S5h-i, we did not detect any difference in the histological analysis of the kidneys obtained at the end of the treatment with water (control) and harmol. Also, we measured the blood urea nitrogen, and did not see any difference between control and harmol-treated mice. Therefore, we consider that harmol did not produce any renal toxicity, at least during the 3 months of treatment.

[REVIEWER]: Figure 5: In vitro experiments showed significant effects of harmol on myotubes but in vivo experiments failed to consider any skeletal muscle protein. Please examine the impacts of harmol aged skeletal muscle translation—for example, measures of myogenin, actinin, or MyHC. Although metabolic panels were provided, this could be misleading because the cause and effects are primarily hidden in whole-tissue assays.

[AUTHORS] We have measured the amount of myosin heavy chain (MHC) in the soleus muscle of harmol-treated mice to compare them with their control group. We have found higher levels of muscle MHC in the group supplemented with harmol (new Figure 7j), indicating improved muscle functionality.

On the other hand, we have studied the amount of MHC in soleus muscle in both harmol- and rosiglitazone-treated, obese and prediabetic young male mice from Figure 5, and compared them with their control vehicle-treated group. In this case, we did not find any differences between these three groups (Figure S5k).

[REVIEWER]: Furthermore, what is the impact of harmol on muscle fiber typing (fast and slow-twitch, muscle cross-sectional area)? Switch of fast to slow-twitch or vice versa may affect metabolism measures. Please examine muscle fibers type in-depth upon treatment with harmol.

[AUTHORS] Following this reviewer's suggestion, we have measured the cross-sectional area of the soleus muscle fibers of harmol-treated old mice and compared it with the control group. We have studied their fiber size and their frequency of distribution (Figure S7d-f), but no differences between both treatment groups were found.

In addition, we have characterized the typology of the musculoskeletal fibers to study the effect of harmol treatment on fast to slow twitch or vice versa. We did not find differences between treatments after following a high-fat diet (Figure S5i). In contrast, in old harmol-treated mice, we have found a higher percentage of type II fibers and a lower percentage of type I fibers compared to the control group (Figure S7g-h).

[REVIEWER]: Figure 6: Please explain how the authors have determined the samples size for the behavioral tests? I believe fewer samples in each sex have made the authors combine the male and female mice for the behavior data?

[AUTHORS] Indeed, gathering 2 year-old mice was a time- and money-consuming process, and in our analysis we put together both sexes. Therefore, sample size was determined by mouse availability. However, there are reports in the literature with roughly similar number of very old mice included, that prompted us to perform the experiments that, ultimately, yielded very clear and significant effects.

[REVIEWER]: Explain why Tukey correction was done for rotarod and grip strength and not for ladder climbing, grid hanging, and running tests?

[AUTHORS] We are sorry for our mistake in indicating the statistical correction tests for multiple comparisons in this figure. Indeed, Tuckey's correction was used for all the experiments with these mice, since the type of data was the same for all of them. We have corrected this typo in the new version.

[REVIEWER]: It is exciting to see the marked difference in grid hanging duration after four months of harmol treatment, but at the same time, it brings the question of how could these 27-month-old (~45 gram) mice hang for more than 5 minutes?

[AUTHORS] In order to eliminate the possible effect of the weight of the mice on the time until falling obtained in the grid-hanging test, we have multiplied the values of time by grams of body mass. Without differences in the weight of the mice between groups (Figure S7a), the increase in performance in the grid-hanging test after the harmol treatment is maintained (Figure S7b).

On the other hand, in the Freshage group in Valencia we have functionally characterized C57BL/6J mice during the aging process at different time points. The oldest mice perform the worst in the grid-hanging test (less than one minute). However, we have seen that after some interventions such as physical exercise, old mice improve their performance, some of them up to more than 5 minutes (Figure 2 for reviewers)

Figure 2 for reviewers. Grid-hanging test in old mice with training.

[REVIEWER]: Remarkable life-extending effects observed in *C. Elegans* worms and *D. melanogaster*. Do you see the effects in telomerase expression or activity?

[AUTHORS] We thank this reviewer for her/his positive comment. We have quantified the expression level of *C. elegans trt-1* gene, encoding for the catalytic subunit of telomerase. As shown in new Figure S6c, treatment of WT worms with 15 µg/ml harmol for 1 week did not affect the expression of *trt-1* mRNA. We only measured this gene in *C. elegans*, since *D. melanogaster* does not count with telomerase genes¹⁹.

[REVIEWER]: Does harmol act directly or break down into metabolites activating observed signaling. Please clarify.

[AUTHORS] This is indeed a very relevant issue. Actually, when we measured the levels of harmol in plasma, brain and liver at different times after harmol administration (Figure 4 and S4), we observed a peak eluting close to the harmol peak. This peak was only present in the harmol-treated liver and plasma samples, but not in the brain samples. This indicated that harmol is metabolized to, at least, one other compound that is not able to cross the BBB. Whether this harmol metabolite is bioactive and may be responsible for some of the mitohormetic properties of harmol, we do not know yet. We are currently studying the nature of this metabolite, and whether there might be other harmol metabolites with relevant bioactivities, although this study escapes the scope of the present work.

[REVIEWER]: Does harmol affect amino acids as it did on glucose and lipids? Amino acids connect metabolism to neuro-activation. Example tryptophan, aspartate, & glutamate have substantial effects on the GABA, AMPK, LC3. Please state your opinions on these straightforward possibilities.

Discuss.

[AUTHORS] This is indeed a very thrilling possibility, since aminoacid metabolism may have a very relevant metabolic impact. Although we have not measured aminoacid metabolism in our harmol-treated samples, we consider this an important field of research for future studies, and have indicated this in the Discussion section.

[REVIEWER]: Please review entire documents for typos, run-ons, and repetitions. In addition, please use punctuation and avoid complex sentences.

[AUTHORS]: We have carefully revised for typos, run-ons and repetitions; and corrected punctuation. We hope the text is clearer now.

Reviewer #4 (Remarks to the Author):

In this study, authors found harmol, a member of the beta-carbolines family with anti-depressant properties, as a novel mitohormetic compound. Harmol induced a transient mitochondrial depolarization, mitophagy and AMPK activation in cultured C2C12 myotubes, in the mouse liver, BAT and muscle. The effects of harmol treatment on mitochondria were mimicked by simultaneous modulation of monoamine-oxidase B and GABAAR receptor. Harmol treatment improved symptoms in diet-induced prediabetic mice. Harmol or a combination of MAO-B and GABAAR modulators extended the lifespan of *Caenorhabditis elegans* or *Drosophila melanogaster*, and delayed frailty onset in mice. They concluded that these results reveal a link between improved psychiatric status and healthspan extension through peripheral mitohormesis.

[REVIEWER]: The effects of harmol are novel and provide mechanistic significant insights into mitohormesis. Their results support the interaction between Harmol and MAO-B and GABAAR to extend healthspan via its mitohormetic action in the peripheral tissues. However, it was not clear how this study revealed a link between psychiatric status and healthspan extension through peripheral mitohormesis (lines 68-70). Although the mitohormetic effect was shown in the peripheral tissues, the effect of harmol treatment on mitochondria in the nervous system is not investigated. Also, how harmol affects on mouse behavior related to anxiety are not shown.

[AUTHORS] This is a very relevant point. To address it, we have performed several tests to define better the effects of harmol in the CNS, and its possible behavior effects, as described in the following answers to this reviewer. Overall, our results indicate that harmol has a very low ability to cross the BBB (Table S1 and Figure 4a-b), exerts a very mild mitohormetic effect in brain (only a non-significant increase in PINK1 levels, Figure 4c), and has a light anxiolytic effect, only apparent upon strong aversive stimuli such as direct illumination of high intensity, as shown in Figure 4d-f. From these results, we can conclude that harmol effects in peripheral tissues is much stronger than in the CNS. Our interpretation of these results is that anxiolytic stimuli that inhibit MAO and GABAAR signaling will elicit a mitohormetic response in peripheral tissues, with all the positive implications for healthspan that we have shown in our work. These anxiolytic, anti-depressant stimuli include harmol, with a limited BBB crossing ability and minor effects in CNS, as we have shown; but they also include other anxiolytic stimuli that, inhibiting MAO-B and GABAAR, will elicit the same mitohormetic response. This is indeed a very exciting field of research that we want to pursue in the future.

Thus, this work does not seem to reveal a link between psychiatric status and peripheral mitohormesis. Detailed comments are listed below.

Major points

[REVIEWER]: 1. Authors showed harmol passed BBB to some extent ($\text{LogPe} > -4.7$, $\text{Ca(t)}/\text{Cd(0)} * 100 = 24.64\%$), or it also affects mitochondrial homeostasis in the nervous system?

[AUTHORS] This is indeed an extremely relevant question. Harmol does cross the BBB, although poorly, as we showed in our PAMPA-BBB assay in Table S1. In addition, we have now observed a brain/plasma AUC ratio of 0.125 in our new kinetic study of harmol

(where values <0.1 indicate impermeability, and values >0.3 indicate sufficient permeability), shown in Figure 4a-b. Therefore, some harmol is present in the CNS after treatment, although much lower than in plasma or liver. To find out if this low amount of harmol affects mitochondrial homeostasis, we analyzed brain samples from mice treated with harmol and sacrificed 7 hours later, when brain concentrations of harmol reach their peak according to Figure 4a. We only detected a mild, non-significant increase in PINK1 levels (Figure 4c), while pAMPK and LC3II-I were not altered by harmol (Figure S4c).

[REVIEWER]: 2. Is there any correlation between mitophagy in the peripheral tissues and psychiatric status after treatment with Harmol?

[AUTHORS] To answer this relevant question, we performed several tests: first, we measured the brain mitophagy response of the same mice from which the liver, BAT and gastrocnemius samples had been obtained in Figure 4, to compare mitophagy between the brain and the peripheral tissues liver, BAT and skeletal muscle. In accordance with the limited ability of harmol to cross the BBB (Table S1 and Figure 4a-b), harmol treatment elicited a very mild effect in the central nervous system: it did not alter the activity of the AMPK-ACC axis, did not affect the autophagy marker LC3-II/LC3-I, and only elicited a light induction of PINK1 in the brain, that did not reach significance (Figure 4c and S4c). Then, we measured the behavioral response of harmol-treated mice in tests that present different levels of aversive stimuli, such as the open field test (Figure 4d), the elevated plus maze test (Figure 4e) and the dark/light box test (Figure 4f). In agreement with our previous results with harmol in the CNS, and also consistent with harmol activity as inhibitor of MAO-B and inverse agonist of the GABAAR, harmol did not have any behavior effect in assays that present mild stressors (open field and elevated plus maze tests), and we only detected a significant anxiolytic effect of harmol in the dark/light box test, involving a strong aversive stimulus.

[REVIEWER]: 3. Figure 4. Harmol and Rosiglitazone show different effects on PINK1 levels in the gastrocnemius: is it expected? Please discuss.

[AUTHORS] Indeed, while PINK1 is increased with both harmol and rosiglitazone treatments in liver and BAT, in gastrocnemius only harmol treatment induced increased levels of PINK1. We have added new WBs of PINK1 in brain samples at 7 hours after treatment, and we also observe increased PINK1 levels (although not significant) with harmol treatment, but not with rosiglitazone, resembling our findings in gastrocnemius. We are not sure of the molecular basis of the different effects of rosiglitazone regarding PINK1 levels in different tissues compared with harmol. Also, we are not aware of other reports studying this phenomenon before. We hypothesize that rosiglitazone effects on mitophagy is not as strong as those coming from harmol, a treatment that consistently induced PINK1 levels in all tested tissues. Rosiglitazone, a PPAR-g agonist, has different targets and mitohormesis-causing mechanisms of action than harmol, and therefore it is not unexpected that their effects are not exactly the same in all tested tissues and conditions.

[REVIEWER]: 4. Figure 6. Control for Harmol was DMSO in a, b, c, and in Figure 6d, S6d,f, and Table S2d,e: control treatment was with EtOH. Figures S6d and 6c, EtOH changed to

DMSO for WT *D. melanogaster*, information on the figure is not consistent, while data is from the same experiment.

[AUTHORS] Indeed, harmol is soluble in water and DMSO. All the experiments using harmol use either DMSO (for invertebrates) or water (for mice) as solvents. In the experiments with selegilin and FG7142 (both of them are soluble in ethanol and DMSO but not in water), we wanted to use harmol as a positive control. For the first experiment with this combination, in *Drosophila*, we initially decided to dissolve all compounds in ethanol as a first approach, since harmol datasheets indicated that all compounds should be soluble in ethanol. Selegiline and FG7142 were both soluble in ethanol and showed very clear lifespan-extending effects, as shown in new Figure 6f-g and Table S3g-h. However, when preparing the food for the flies, we could not solubilize harmol in ethanol, and therefore we could not include this control in these experiments (Figure 6f-g). For the next experiments with *C. elegans*, we decided to use the other solvent available for selegilin, FG7142 and harmol: DMSO, used in Figure 6d-e. In these experiments, we validated the lifespan extension ability of harmol, but the combination of selegilin+FG7142 did not extend lifespan (Figure S6j). We understand that this change in vehicle might be somewhat disorienting. Still, we consider that all proper controls are always included, and therefore the conclusions from each experiment are robust.

In addition, there was a typo in previous Figure 6c (present Figure 6f), where DMSO was indicated as the solvent. Actually, in this experiment we used ethanol as solvent, as indicated in previous Table S2d and e (now Table S3g-h) where ethanol was correctly indicated as the solvent. We have corrected this typo in the new version, Figure 6f.

[REVIEWER]: 5. Lines 555-557: It seems to be a wrong statement. According to data in table S1 table, harmine is totally impermeable $\log P_{e} < -4.7$, but harmol is not. So, it is not clear how it is affected CNS.

[AUTHORS]: We have corrected this mistake in the new manuscript text, and updated the information about the ability of harmol to cross the BBB and affect brain mitophagy and anxiogenic stress response.

Minor comments:

[REVIEWER]: 1. Line 352: Possibly “-“ was missed “($\log P_{e} < -4.7$)”, because it was stated that $\log P_{e} < -4.7$ refers to a totally impermeable compound

[AUTHORS]: We have corrected this typo.

[REVIEWER]: 2. Line 458: S1a -> S2a; Line 462: S1b -> S2b and further, S1 table was BBB permeability table.

[AUTHORS]: We have corrected these mistakes in the new text.

3. Misspellings: Line 697: carried our -> carried out, line 196: ith -> with.

[AUTHORS]: We have corrected these misspellings.

- 1 Pardo R, Enguix N, Lasheras J, Feliu JE, Kralli A, Villena JA. Rosiglitazone-induced mitochondrial biogenesis in white adipose tissue is independent of peroxisome proliferator-activated receptor γ coactivator-1 α . *PLoS One* 2011; **6**. doi:10.1371/JOURNAL.PONE.0026989.

- 2 Chaput E, Saladin R, Silvestre M, Edgar AD. Fenofibrate and Rosiglitazone Lower Serum Triglycerides with Opposing Effects on Body Weight. *Biochem Biophys Res Commun* 2000; **271**: 445–450.
- 3 Fryer LGD, Parbu-Patel A, Carling D. The anti-diabetic drugs rosiglitazone and metformin stimulate AMP-activated protein kinase through distinct signaling pathways. *Journal of Biological Chemistry* 2002; **277**: 25226–25232.
- 4 Apfeld J, O'Connor G, McDonagh T, DiStefano PS, Curtis R. The AMP-activated protein kinase AAK-2 links energy levels and insulin-like signals to lifespan in *C. elegans*. *Genes Dev* 2004; **18**: 3004–3009.
- 5 Knoll J, Miklya I. Longevity study with low doses of selegiline/(-)-deprenyl and (2R)-1-(1-benzofuran-2-yl)-N-propylpentane-2-amine (BPAP). *Life Sci* 2016; **167**: 32–38.
- 6 Roelofs J, van Haastert PJM. Genes lost during evolution. *Nature* 2001 **411**:6841 2001; **411**: 1013–1014.
- 7 Dewhurst SA, Croker SG, Ikeda K, McCaman RE. Metabolism of biogenic amines in drosophila nervous tissue. *Comparative Biochemistry and Physiology Part B: Comparative Biochemistry* 1972; **43**: 975–981.
- 8 Chaudhuri A, Bowling K, Funderburk C, Lawal H, Inamdar A, Wang Z *et al*. Interaction of Genetic and Environmental Factors in a Drosophila Parkinsonism Model. *Journal of Neuroscience* 2007; **27**: 2457–2467.
- 9 Wang Z, Ferdousy F, Lawal H, Huang Z, Daigle JG, Izevbaye I *et al*. Catecholamines up integrates dopamine synthesis and synaptic trafficking. *J Neurochem* 2011; **119**: 1294–1305.
- 10 Yellman C, Tao H, He B, Hirsh J. Conserved and sexually dimorphic behavioral responses to biogenic amines in decapitated Drosophila. *Proc Natl Acad Sci U S A* 1997; **94**: 4131–4136.
- 11 Martin CA, Krantz DE. Drosophila melanogaster as a genetic model system to study neurotransmitter transporters. *Neurochem Int* 2014; **73**: 71.
- 12 Seifert A, Allan LA, Clarke PR. DYRK1A phosphorylates caspase 9 at an inhibitory site and is potently inhibited in human cells by harmine. *FEBS J* 2008; **275**: 6268–6280.
- 13 Bain J, Plater L, Elliott M, Shpiro N, Hastie CJ, Mclauchlan H *et al*. The selectivity of protein kinase inhibitors: a further update. *Biochem J* 2007; **408**: 297.
- 14 Tarpley M, Oladapo HO, Strepay D, Caligan TB, Chdid L, Shehata H *et al*. Identification of harmine and β -carboline analogs from a high-throughput screen of an approved drug collection; profiling as differential inhibitors of DYRK1A and monoamine oxidase A and for in vitro and in vivo anti-cancer studies. *Eur J Pharm Sci* 2021; **162**. doi:10.1016/J.EJPS.2021.105821.
- 15 Corona JC, Duchon MR. PPAR γ as a therapeutic target to rescue mitochondrial function in neurological disease. *Free Radic Biol Med* 2016; **100**: 153–163.
- 16 Rong JX, Klein JLD, Qiu Y, Xie M, Johnson JH, Waters KM *et al*. Rosiglitazone Induces Mitochondrial Biogenesis in Differentiated Murine 3T3-L1 and C3H/10T1/2 Adipocytes. *PPAR Res* 2011; **2011**: 11.
- 17 Perry SW, Norman JP, Barbieri J, Brown EB, Gelbard HA. Mitochondrial membrane potential probes and the proton gradient: a practical usage guide. *Biotechniques* 2011; **50**: 98–115.

- 18 Mathur A, Hong Y, Kemp BK, Barrientos AA, Erusalimsky JD. Evaluation of fluorescent dyes for the detection of mitochondrial membrane potential changes in cultured cardiomyocytes. *Cardiovasc Res* 2000; **46**: 126–138.
- 19 Pardue M-L, DeBaryshe GP. Drosophila Telomeres: A Variation on the Telomerase Theme. 2013.<https://www.ncbi.nlm.nih.gov/books/NBK6617/> (accessed 21 Nov2022).

REVIEWER COMMENTS

Reviewer #1 (Remarks to the Author):

The revised version of the manuscript has been significantly improved, also by implementing additional experimental data suggested by the reviewers, including myself.

I have two questions regarding the newly implemented experimental data:

(i) new Fig. 6e: In the triple mutant of AMX, harmol still increases lifespan, however to a reduced extent (18.8% compared to 37.55 % in WT/N2).

The authors interpret this as an indication for MAOs/AMXs indeed serving as targets for harmol.

It is unclear to me how the *absence* of a target can mediate lifespan extension by its proposed inhibitor.

To me this rather indicates that additional targets (i.e. besides MAO/AMX) exist, and that these targets mediate approx 50% of the lifespan extension in *C. elegans*.

(On a side note, the authors interpret their new findings in Fig. 6c by analogy)

(ii) new Figure S6b / basal AMPK 2532S: To my knowledge (and as stated in my initial Comments for Authors), the Cell Signaling 2532S basal AMPK antibody does *not* work in nematodal lysates, and other *C. elegans* labs agree on this notion. The amino acid sequence of nematodal AMPK is very different from mammalian (while the Thr172 phospho site is well conserved).

Are the authors *sure* they have detected basal AMPK (rather than an unspecific signal)?

The typical approach to resolve this would be testing the antibody on lysates from WT/N2 next to lysates from *aak-2*-deficient nematodes.

Reviewer #2 (Remarks to the Author):

The authors have addressed my questions satisfactorily and I have no further questions and recommend acceptance.

Reviewer #4 (Remarks to the Author):

This work revealed a correlation between the effect of harmol on mitohormesis in the peripheral tissues and the life span extension. However, the authors did not address my primary concern that this work does not reveal a link between psychiatric status and peripheral mitohormesis. The authors' replies further support that harmol works in peripheral tissues without significant psychiatric effects.

The authors state, 'Despite mounting evidence, the molecular link between metabolism and psychiatric

status is not well understood' (line 71) and 'Psychiatric and psychologic status have long been associated with healthspan, with mental disorders correlating with shorter lifespan, and higher wellbeing scores associated with longer lifespan^{1,2}. However, the molecular mechanisms responsible for these associations are not yet understood.' (line 90-). However, this work does not provide data that directly addresses these questions. I suggest rewriting the abstract and introduction for better logical flow and consistency.

POINT-BY-POINT RESPONSE TO REVIEWER COMMENTS

We heartfully thank the reviewers for their time and effort revising our manuscript, and for their very positive comments.

Reviewer #1:

[REVIEWER]: The revised version of the manuscript has been significantly improved, also by implementing additional experimental data suggested by the reviewers, including myself.

[AUTHORS]: We thank this reviewer for this positive evaluation of our work.

[REVIEWER]: I have two questions regarding the newly implemented experimental data: (i) new Fig. 6e: In the triple mutant of AMX, harmol still increases lifespan, however to a reduced extent (18.8% compared to 37.55 % in WT/N2).

The authors interpret this as an indication for MAOs/AMXs indeed serving as targets for harmol.

It is unclear to me how the *absence* of a target can mediate lifespan extension by its proposed inhibitor.

To me this rather indicates that additional targets (i.e. besides MAO/AMX) exist, and that these targets mediate approx 50% of the lifespan extension in *C. elegans*.

(On a side note, the authors interpret their new findings in Fig. 6c by analogy).

[AUTHORS]: We agree completely with this comment. Indeed, our model implies that harmol benefits on lifespan and healthspan extension require its simultaneous effects on MAO-B (inhibiting its function, as rasagilin and selegilin) and on GABAAR (as an inverse agonist, as FG-7142). Acting only on one of these targets (rasagilin or FG-7142 in separate) does not recapitulate harmol effects (see Figure 3i, j; Figure S3e). Therefore, in the amx-1/2/3 mutant worms, harmol effects on MAO will be achieved by default by the amx mutations themselves, and harmol will not need to inhibit the already absent MAO anymore. At the same time, harmol will still be acting as a GABAAR inverse agonist, and this effect, together with the absence of MAO activity, will elicit mitohormesis and lifespan extension, as shown in new Fig. 6e.

[REVIEWER]: (ii) new Figure S6b / basal AMPK 2532S: To my knowledge (and as stated in my initial Comments for Authors), the Cell Signaling 2532S basal AMPK antibody does *not* work in nematodal lysates, and other *C. elegans* labs agree on this notion. The amino acid sequence of nematodal AMPK is very different from mammalian (while the Thr172 phospho site is well conserved).

Are the authors *sure* they have detected basal AMPK (rather than an unspecific signal)?

The typical approach to resolve this would be testing the antibody on lysates from WT/N2 next to lysates from aak-2-deficient nematodes.

[AUTHORS]: We thank this reviewer for his/her comment. We did not understand her/his initial comment, where the reviewer indicated that the Cell Signaling antibody against the non-phosphorylated form of AMPK was not appropriate. Actually, we were sure that we were following his/her advice when performing the WB against tAMPK and pAMPK shown in Figure S6b. As shown in the Source Data, the band that appeared above

the 50 KDa mark in our WB against tAMPK was as strong and consistent as that observed in the WB against pAMPK, although slightly lower, as often happens with antibodies against unphosphorylated forms of a protein. Reviewing the available commercial information, we can see that neither the pAMPK (40H9) nor the tAMPK (2532S) antibodies are expected to react against the *C. elegans* forms of the protein. We tried blotting against tubulin, as suggested by this reviewer, using our mouse and human-specific antibodies. Regrettably, we did not get any signal and did not have any remaining sample to try different antibodies.

We agree with the reviewer that the best way of determining the specificity of an antibody is to check if it recognizes the protein in samples from WT individuals, and that this recognition disappears in the KO mutant. However, we have not been able to gather protein lysates from aak2 mutant worms. If strictly necessary, we may try to obtain these samples, but it would be problematic for us in this case.

We also humbly consider that, since these antibodies are polyclonal, we cannot exclude that different antibody lots used by different laboratories may show different reactivities against isoforms from species as distant from the human peptide used for the antibody generation as *C. elegans*. Therefore, we postulate that our results are correct and may differ from those obtained with different lots of these polyclonal antibodies elsewhere.

Reviewer #2 (Remarks to the Author):

[REVIEWER]: The authors have addressed my questions satisfactorily and I have no further questions and recommend acceptance.

[AUTHORS]: We thank this reviewer for her/his work and help improving our article.

Reviewer #1 as substitute of Reviewer #3:

[REVIEWER]: It should be clarified whether the quantification of the Western blots in figure 7j were normalized to beta-actin in both the figure (y-axis) and the legend.

[AUTHORS]: Thanks to this comment, we have identified a small mistake in this panel: actually, as suggested by this reviewer, band intensity quantifications were normalized by actin, and not by Water control. We have corrected this mistake in the new figure panel and legend.

Reviewer #4 (Remarks to the Author):

[REVIEWER]: This work revealed a correlation between the effect of harmol on mitohormesis in the peripheral tissues and the life span extension. However, the authors did not address my primary concern that this work does not reveal a link between psychiatric status and peripheral mitohormesis. The authors' replies further support that harmol works in peripheral tissues without significant psychiatric effects.

The authors state, 'Despite mounting evidence, the molecular link between metabolism and psychiatric status is not well understood' (line 71) and 'Psychiatric and psychologic status have long been associated with healthspan, with mental disorders correlating with shorter lifespan, and higher wellbeing scores associated with longer lifespan^{1,2}. However, the molecular mechanisms responsible for these associations are not yet understood.' (line 90-). However, this work does not provide data that directly addresses

these questions. I suggest rewriting the abstract and introduction for better logical flow and consistency.

[AUTHORS]: We agree with this reviewer comment, and have rewritten our abstract and introduction to focus on the peripheral effects of harmol. We do see positive effects in mitochondrial function (Figure 3j), and fly lifespan (Figure 6f, g) after a treatment with a combination of a MAO inhibitor and a GABAAR inverse agonist, that indicate that this combination of anti-depressant and anxiolytic drugs can reproduce harmol benefits. However, we agree with this reviewer that our work as a whole is not focused on these drug effects, and stressing them in the abstract and introduction could be misleading. Therefore, we have rewritten these sections and focused on the mitohormetic and peripheral effects of harmol in our models. We hope this new version is clearer now.

REVIEWERS' COMMENTS

Reviewer #1 (Remarks to the Author):

The remaining issue appears to be the reviewer's question re Suppl. Fig. S6b, whether the CellSignaling tAMPK (2532S) antibody used is capable of detecting *C. elegans* total AMPK (which, as stated in the first round of reviews, is not the case, at least in the hands of the reviewer, and others).

While this tAMPK antibody is polyclonal (which might explain different outcomes in different labs), the pAMPK antibody used is not (as stated on the website of the manufacturer, and unlike stated by the authors).

However, and since at least the phosphorylated signal is mechanistically in line with the *aak2*-mutant lifespans (Suppl. Fig. S6g), the reviewer refrains from repeating the demand for an tAMPK blot of *aak2* samples, despite the fact that these samples would be easy to obtain (unlike stated by the authors).

POINT-BY-POINT RESPONSE TO REVIEWER COMMENTS

We heartfully thank the reviewers for their time and effort revising our manuscript, and for their very positive comments.

Reviewer #1:

[REVIEWER]: The remaining issue appears to be the reviewer's question re Suppl. Fig. S6b, whether the CellSignaling tAMPK (2532S) antibody used is capable of detecting *C. elegans* total AMPK (which, as stated in the first round of reviews, is not the case, at least in the hands of the reviewer, and others).

While this tAMPK antibody is polyclonal (which might explain different outcomes in different labs), the pAMPK antibody used is not (as stated on the website of the manufacturer, and unlike stated by the authors).

However, and since at least the phosphorylated signal is mechanistically in line with the *aak2*-mutant lifespans (Suppl. Fig. S6g), the reviewer refrains from repeating the demand for an tAMPK blot of *aak2* samples, despite the fact that these samples would be easy to obtain (unlike stated by the authors).

[AUTHORS]: We heartfully thank this reviewer for his time and insight, that have greatly helped improving our manuscript. We also thank him for his understanding with regards to Supplementary Figure S6g. We agree with him that our results with pAMPK are robust enough and in line with the observed phenotype.